# Learning Robust Representations via Multi-View Information Bottleneck

**Marco Federici**
University of Amsterdam
m.federici@uva.nl

**Anjan Dutta**
University of Exeter
a.dutta@exeter.ac.uk

**Patrick Forré**
University of Amsterdam
p.d.forre@uva.nl

**Nate Kushmann**
Microsoft Research
nkushman@microsoft.com

**Zeynep Akata**
University of Tuebingen
zeynep.akata@uni-tuebingen.de

## Abstract

The information bottleneck principle provides an information-theoretic method for representation learning, by training an encoder to retain all information which is relevant for predicting the label while minimizing the amount of other, excess information in the representation. The original formulation, however, requires labeled data to identify the superfluous information. In this work, we extend this ability to the multi-view unsupervised setting, where two views of the same underlying entity are provided but the label is unknown. This enables us to identify superfluous information as that not shared by both views. A theoretical analysis leads to the definition of a new multi-view model that produces state-of-the-art results on the Sketchy dataset and label-limited versions of the MIR-Flickr dataset. We also extend our theory to the single-view setting by taking advantage of standard data augmentation techniques, empirically showing better generalization capabilities when compared to common unsupervised approaches for representation learning.

## 1 Introduction

The goal of deep representation learning (LeCun et al., 2015) is to transform a raw observational input, $\mathbf{x}$, into a, typically lower-dimensional, representation, $\mathbf{z}$, that contains the information relevant for a given task or set of tasks. Significant progress has been made in deep learning via supervised representation learning, where the labels, $\mathbf{y}$, for the downstream task are known while $p(\mathbf{y}|\mathbf{x})$ is learned directly (Sutskever et al., 2012; Hinton et al., 2012). Due to the cost of acquiring large labeled datasets, a recently renewed focus on unsupervised representation learning seeks to generate representations, $\mathbf{z}$, that are useful for a wide variety of different tasks where little to no labeled data is available (Devlin et al., 2018; Radford et al., 2019).

Our work is based on the information bottleneck principle (Tishby et al., 2000) where a representation becomes less affected by nuisances by discarding all information from the input that is not useful for a given task, resulting in increased robustness. In the supervised setting, one can directly apply the information bottleneck principle by minimizing the mutual information between the data $\mathbf{x}$ and its representation $\mathbf{z}$, $I(\mathbf{x}; \mathbf{z})$, while simultaneously maximizing the mutual information between $\mathbf{z}$ and the label $\mathbf{y}$ (Alemi et al., 2017). In the unsupervised setting, discarding only superfluous information is more challenging, as without labels the model cannot directly identify which information is relevant. Recent literature (Devon Hjelm et al., 2019; van den Oord et al., 2018) has focused on the InfoMax objective *maximizing $I(\mathbf{x}, \mathbf{z})$* instead of minimizing it, to guarantee that all the predictive information is retained by the representation, but doing nothing to discard the irrelevant information.

In this paper, we extend the information bottleneck method to the unsupervised multi-view setting. To do this, we rely on a basic assumption of the multi-view literature – that each view provides the same *task-relevant information* (Zhao et al., 2017). Hence, one can improve generalization by discarding all the information not shared by both views from the representation. We do this by maximizing the mutual information between the representations of the two views (Multi-View

InfoMax objective) while at the same time eliminating the information not shared between them, since it is guaranteed to be superfluous. The resulting representations are more robust for the given task as they have eliminated view specific nuisances.

Our contributions are three-fold: (1) We extend the information bottleneck principle to the unsupervised multi-view setting and provide a rigorous theoretical analysis of its application. (2) We define a new model [1] that empirically leads to state-of-the-art results in the low-label setting on two standard multi-view datasets, Sketchy and MIR-Flickr. (3) By exploiting data augmentation techniques, we empirically show that the representations learned by our model in single-view settings are more robust than existing unsupervised representation learning methods, connecting our theory to the choice of augmentation strategy.

## 2 PRELIMINARIES AND FRAMEWORK

The challenge of representation learning can be formulated as finding a distribution $p(\mathbf{z}|\mathbf{x})$ that maps data observations $\mathbf{x} \in \mathbb{X}$ into a representation $\mathbf{z} \in \mathbb{Z}$, capturing some desired characteristics. Whenever the end goal involves predicting a label $\mathbf{y}$, we consider only $\mathbf{z}$ that are discriminative enough to identify $\mathbf{y}$. This requirement can be quantified by considering the amount of label information that remains accessible after encoding the data, and is known as sufficiency of $\mathbf{z}$ for $\mathbf{y}$ (Achille & Soatto, 2018):

**Definition 1.** Sufficiency: A representation $\mathbf{z}$ of $\mathbf{x}$ is sufficient for $\mathbf{y}$ if and only if $I(\mathbf{x}; \mathbf{y}|\mathbf{z}) = 0$.

Any model that has access to a sufficient representation $\mathbf{z}$ must be able to predict $\mathbf{y}$ at least as accurately as if it has access to the original data $\mathbf{x}$ instead. In fact, $\mathbf{z}$ is sufficient for $\mathbf{y}$ if and only if the amount of information regarding the task is unchanged by the encoding procedure (see Proposition B.1 in the Appendix):

$$I(\mathbf{x}; \mathbf{y}|\mathbf{z}) = 0 \iff I(\mathbf{x}; \mathbf{y}) = I(\mathbf{y}; \mathbf{z}). \tag{1}$$

Among sufficient representations, the ones that result in better generalization for unlabeled data instances are particularly appealing. When $\mathbf{x}$ has higher information content than $\mathbf{y}$, some of the information in $\mathbf{x}$ must be irrelevant for the prediction task. This can be better understood by subdividing $I(\mathbf{x}; \mathbf{z})$ into two components by using the chain rule of mutual information (see Appendix A):

$$I(\mathbf{x}; \mathbf{z}) = \underbrace{I(\mathbf{x}; \mathbf{z}|\mathbf{y})}_{\text{superfluous information}} + \underbrace{I(\mathbf{y}; \mathbf{z})}_{\text{predictive information}}. \tag{2}$$

Conditional mutual information $I(\mathbf{x}; \mathbf{z}|\mathbf{y})$ represents the information in $\mathbf{z}$ that is not predictive of $\mathbf{y}$, i.e. **superfluous information**. While $I(\mathbf{y}; \mathbf{z})$ determines how much label information is accessible from the representation. Note that this last term is independent of the representation as long as $\mathbf{z}$ is sufficient for $\mathbf{y}$ (see Equation 1). As a consequence, a sufficient representation contains minimal data information whenever $I(\mathbf{x}; \mathbf{z}|\mathbf{y})$ is minimized.

Minimizing the amount of superfluous information can be done directly only in supervised settings. In fact, reducing $I(\mathbf{x}; \mathbf{z})$ without violating the sufficiency constraint necessarily requires making some additional assumptions on the predictive task (see Theorem B.1 in the Appendix). In the next section we describe the basis of our technique, a strategy to safely reduce the information content of a representation even when the label $\mathbf{y}$ is not observed, by exploiting redundant information in the form of an additional view on the data.

## 3 MULTI-VIEW INFORMATION BOTTLENECK

As a motivating example, consider $\mathbf{v}_1$ and $\mathbf{v}_2$ to be two images of the same object from different view-points and let $\mathbf{y}$ be its label. Assuming that the object is clearly distinguishable from both $\mathbf{v}_1$ and let $\mathbf{v}_2$, any representation $\mathbf{z}$ containing all information accessible from both views would also contain the necessary label information. Furthermore, if $\mathbf{z}$ captures only the details that are visible from both pictures, it would eliminate the view-specific details and reduce the sensitivity of the representation to view-changes. The theory to support this intuition is described in the following where $\mathbf{v}_1$ and $\mathbf{v}_2$ are jointly observed and referred to as data-views.

---

[1]Code available at https://github.com/mfederici/Multi-View-Information-Bottleneck

## 3.1 Sufficiency and Robustness in the Multi-view setting

In this section we extend our analysis of sufficiency and minimality to the multi-view setting.

Intuitively, we can guarantee that $\mathbf{z}$ is sufficient for predicting $\mathbf{y}$ even without knowing $\mathbf{y}$ by ensuring that $\mathbf{z}$ maintains all information which is shared by $\mathbf{v}_1$ and $\mathbf{v}_2$. This intuition relies on a basic assumption of the multi-view environment – that the two views provide the same predictive information. To formalize this we define **redundancy**.

**Definition 2.** Redundancy: $\mathbf{v}_1$ is redundant with respect to $\mathbf{v}_2$ for $\mathbf{y}$ if and only if $I(\mathbf{y}; \mathbf{v}_1|\mathbf{v}_2) = 0$

Intuitively, a view $\mathbf{v}_1$ is redundant for a task whenever it is irrelevant for the prediction of $\mathbf{y}$ if $\mathbf{v}_2$ is already observed. Whenever $\mathbf{v}_1$ and $\mathbf{v}_2$ are **mutually redundant** ($\mathbf{v}_1$ is redundant with respect to $\mathbf{v}_2$ for $\mathbf{y}$, and vice-versa), we can show the following:

**Corollary 1.** *Let $\mathbf{v}_1$ and $\mathbf{v}_2$ be two mutually redundant views for a target $\mathbf{y}$ and let $\mathbf{z}_1$ be a representation of $\mathbf{v}_1$. If $\mathbf{z}_1$ is sufficient for $\mathbf{v}_2$ ($I(\mathbf{v}_1; \mathbf{v}_2|\mathbf{z}_1) = 0$) then $\mathbf{z}_1$ is as predictive for $\mathbf{y}$ as the joint observation of the two views ($I(\mathbf{v}_1\mathbf{v}_2; \mathbf{y}) = I(\mathbf{y}; \mathbf{z}_1)$).*

In other words, whenever it is possible to assume mutual redundancy, any representation which contains all the information shared by both views (the redundant information) is as predictive as their joint observation.

By factorizing the mutual information between $\mathbf{v}_1$ and $\mathbf{z}_1$ analogously to Equation 2, we can identify two components:

$$I(\mathbf{v}_1; \mathbf{z}_1) = \underbrace{I(\mathbf{v}_1; \mathbf{z}_1|\mathbf{v}_2)}_{\text{superfluous information}} + \underbrace{I(\mathbf{v}_2; \mathbf{z}_1)}_{\text{predictive information for } \mathbf{v}_2} .$$

Since $I(\mathbf{v}_2; \mathbf{z}_1)$ has to be maximal if we want the representation to be sufficient for the label, we conclude that $I(\mathbf{v}_1; \mathbf{z}_1)$ can be reduced by minimizing $I(\mathbf{v}_1; \mathbf{z}_1|\mathbf{v}_2)$. This term intuitively represents the information $\mathbf{z}_1$ contains which is unique to $\mathbf{v}_1$ and is not predictable by observing $\mathbf{v}_2$. Since we assumed mutual redundancy between the two views, this information must be irrelevant for the predictive task and, therefore, it can be safely discarded. The proofs and formal assertions for the above statements and Corollary 1 can be found in Appendix B.

The less the two views have in common, the more $I(\mathbf{v}_1; \mathbf{z}_1)$ can be reduced without violating sufficiency for the label, and consequently, the more robust the resulting representation. At the extreme, $\mathbf{v}_1$ and $\mathbf{v}_2$ share only label information, in which case we can show that $\mathbf{z}_1$ is minimal for $\mathbf{y}$ and our method is identical to the supervised information bottleneck method without needing to access the labels. Conversely, if $\mathbf{v}_1$ and $\mathbf{v}_2$ are identical, then our method degenerates to the InfoMax principle since no information can be safely discarded (see Appendix E).

## 3.2 The Multi-View Information Bottleneck Loss Function

Given $\mathbf{v}_1$ and $\mathbf{v}_2$ that satisfy the mutual redundancy condition for a label $\mathbf{y}$, we would like to define an objective function for the representation $\mathbf{z}_1$ of $\mathbf{v}_1$ that discards as much information as possible without losing any label information. In Section 3.1 we showed that we can obtain sufficiency for $\mathbf{y}$ by ensuring that the representation $\mathbf{z}_1$ of $\mathbf{v}_1$ is sufficient for $\mathbf{v}_2$, and that decreasing $I(\mathbf{z}_1; \mathbf{v}_1|\mathbf{v}_2)$ will increase the robustness of the representation by discarding irrelevant information. So we can combine these two requirements using a relaxed Lagrangian objective to obtain the minimal sufficient representation $\mathbf{z}_1$ for $\mathbf{v}_2$:

$$\mathcal{L}_1(\theta; \lambda_1) = I_\theta(\mathbf{z}_1; \mathbf{v}_1|\mathbf{v}_2) - \lambda_1 \, I_\theta(\mathbf{v}_2; \mathbf{z}_1), \tag{3}$$

where $\theta$ denotes the dependency on the parameters of the encoder $p_\theta(\mathbf{z}_1|\mathbf{v}_1)$, and $\lambda_1$ represents the Lagrangian multiplier introduced by the constrained optimization. Symmetrically, we define a loss $\mathcal{L}_2$ to optimize the parameters $\psi$ of a conditional distribution $p_\psi(\mathbf{z}_2|\mathbf{v}_2)$ that defines a minimal sufficient representation $\mathbf{z}_2$ of the second view $\mathbf{v}_2$ for $\mathbf{v}_1$:

$$\mathcal{L}_2(\psi; \lambda_2) = I_\psi(\mathbf{z}_2; \mathbf{v}_2|\mathbf{v}_1) - \lambda_2 \, I_\psi(\mathbf{v}_1; \mathbf{z}_2), \tag{4}$$

By defining $\mathbf{z}_1$ and $\mathbf{z}_2$ on the same domain $\mathbb{Z}$ and re-parametrizing the Lagrangian multipliers, the average of the two loss functions $\mathcal{L}_1$ and $\mathcal{L}_2$ can be upper bounded as follows:

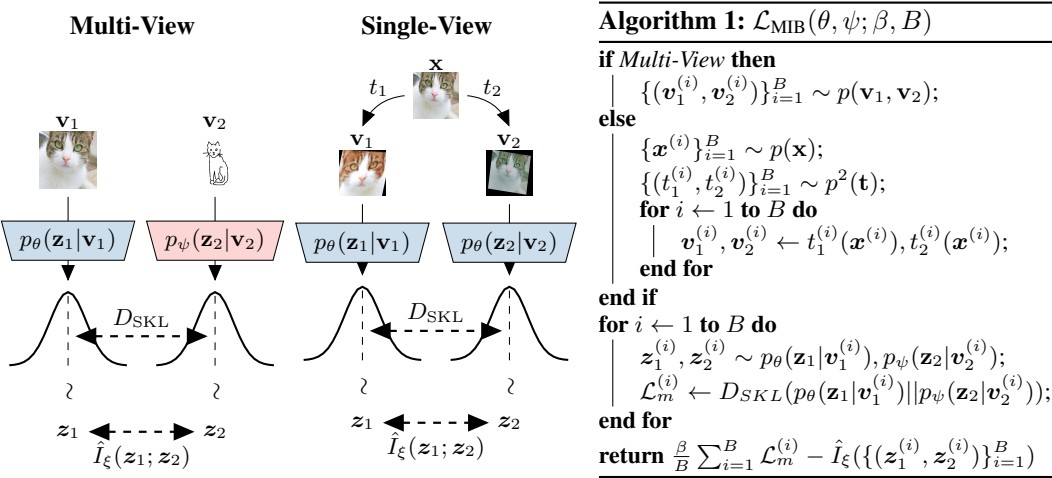

Figure 1: Visualization our Multi-View Information Bottleneck model for both multi-view and single-view settings, where $\hat{I}_\xi(\boldsymbol{z}_1; \boldsymbol{z}_2)$ refers to the sample-based parametric mutual information estimation. Whenever $p(\mathbf{v}_1)$ and $p(\mathbf{v}_2)$ have the same distribution, the two encoders can share their parameters.

$$\mathcal{L}_{MIB}(\theta, \psi; \beta) = -I_{\theta\psi}(\mathbf{z}_1; \mathbf{z}_2) + \beta \, D_{SKL}(p_\theta(\mathbf{z}_1|\mathbf{v}_1)||p_\psi(\mathbf{z}_2|\mathbf{v}_2)), \tag{5}$$

where $D_{SKL}$ represents the symmetrized KL divergence obtained by averaging the expected value of $D_{\mathrm{KL}}(p_\theta(\mathbf{z}_1|\mathbf{v}_1)||p_\psi(\mathbf{z}_2|\mathbf{v}_2))$ and $D_{\mathrm{KL}}(p_\psi(\mathbf{z}_2|\mathbf{v}_2)||p_\theta(\mathbf{z}_1|\mathbf{v}_1))$ for joint observations of the two views, while the coefficient $\beta$ defines the trade-off between sufficiency and robustness of the representation, which is a hyper-parameter in this work. The resulting Multi-View Infomation Bottleneck (MIB) model (Equation 5) is visualized in Figure 1, while the batch-based computation of the loss function is summarized in Algorithm 1.

The symmetrized KL divergence $D_{SKL}(p_\theta(\mathbf{z}_1|\mathbf{v}_1)||p_\psi(\mathbf{z}_2|\mathbf{v}_2))$ can be computed directly whenever $p_\theta(\mathbf{z}_1|\mathbf{v}_1)$ and $p_\psi(\mathbf{z}_2|\mathbf{v}_2)$ have a known density, while the mutual information between the two representations $I_{\theta\psi}(\mathbf{z}_1; \mathbf{z}_2)$ can be maximized by using any sample-based differentiable mutual information lower bound. We tried the Jensen-Shannon $I_{\mathrm{JS}}$ (Devon Hjelm et al., 2019; Poole et al., 2019) and the InfoNCE $I_{\mathrm{NCE}}$ (van den Oord et al., 2018) estimators. These both require introducing an auxiliary parameteric model $C_\xi(\boldsymbol{z}_1, \boldsymbol{z}_2)$ which is jointly optimized during the training procedure using re-parametrized samples from $p_\theta(\mathbf{z}_1|\mathbf{v}_1)$ and $p_\psi(\mathbf{z}_2|\mathbf{v}_2)$. The full derivation for the MIB loss function can be found in Appendix F.

### 3.3 SELF-SUPERVISION AND INVARIANCE

Our method can also be applied when multiple views are not available by taking advantage of standard data augmentation techniques. This allows learning invariances directly from the augmented data, rather than requiring them to be built into the model architecture.

By picking a class $\mathbb{T}$ of data augmentation functions $t : \mathbb{X} \to \mathbb{W}$ that do not affect label information, it is possible to artificially build views that satisfy mutual redundancy for $\mathbf{y}$. Let $\mathbf{t}_1$ and $\mathbf{t}_2$ be two random variables over $\mathbb{T}$, then $\mathbf{v}_1 := \mathbf{t}_1(\mathbf{x})$ and $\mathbf{v}_2 := \mathbf{t}_2(\mathbf{x})$ must be mutually redundant for $\mathbf{y}$. Since data augmentation functions in $\mathbb{T}$ do not affect label information ($I(\mathbf{v}_1; \mathbf{y}) = I(\mathbf{v}_2; \mathbf{y}) = I(\mathbf{x}; \mathbf{y})$), a representation $\mathbf{z}_1$ of $\mathbf{v}_1$ that is sufficient for $\mathbf{v}_2$ must contain same amount of predictive information as $\mathbf{x}$. Formal proofs for this statement can be found in Appendix B.4.

Whenever the two transformations for the same observation are independent ($I(\mathbf{t}_1; \mathbf{t}_2|\mathbf{x}) = 0$), they introduce uncorrelated variations in the two views, which will be discarded when creating a

representation using our training objective. As an example, if $\mathbb{T}$ represents a set of small translations, the two resulting views will differ by a small shift. Since this information is not shared, any $\mathbf{z}_1$ which is optimal according to the MIB objective must discard fine-grained details regarding the position.

To enable parameter sharing between the encoders, we generate the two views $\mathbf{v}_1$ and $\mathbf{v}_2$ by independently sampling two functions from the same function class $\mathbb{T}$ with uniform probability. As a result, $\mathbf{t}_1$ and $\mathbf{t}_2$ will have the same distribution, and so the two generated views will also have the same marginals ($p(\mathbf{v}_1) = p(\mathbf{v}_2)$). For this reason, the two conditional distributions $p_\theta(\mathbf{z}_1|\mathbf{v}_1)$ and $p_\psi(\mathbf{z}_2|\mathbf{v}_2)$ can share their parameters and only one encoder is necessary. Full (or partial) parameter sharing can be also applied in the multi-view settings whenever the two views have the same (or similar) marginal distributions.

## 4 RELATED WORK

The relationship between our method and past work on representation learning is best described using the *Information Plane* (Tishby et al., 2000). In this setting, each representation $\mathbf{z}$ of $\mathbf{x}$ for a predictive task $\mathbf{y}$ can be characterised by the amount of information regarding the raw observation $I(\mathbf{x};\mathbf{z})$ and the corresponding measure of accessible predictive information $I(\mathbf{y};\mathbf{z})$ ($x$ and $y$ axis respectively on Figure 2). Ideally, a good representation would be maximally informative about the label while retaining a minimal amount of information from the observations (top left corner of the parallelogram). Further details on the Information Plane and the bounds visualized in Figure 2 are described in Appendix C.

Thanks to recent progress in mutual information estimation (Nguyen et al., 2008; Ishmael Belghazi et al., 2018; Poole et al., 2019), the InfoMax principle (Linsker, 1988) has gained attention for unsupervised representation learning (Devon Hjelm et al., 2019; van den Oord et al., 2018). Since the InfoMax objective involves maximizing $I(\mathbf{x};\mathbf{z})$, the resulting representation aims to preserve all the information regarding the raw observations (top right corner in Figure 2).

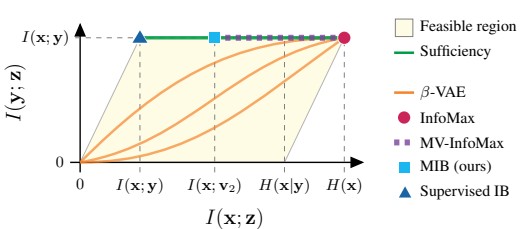

Figure 2: Information Plane determined by $I(\mathbf{x};\mathbf{z})$ (x-axis) and $I(\mathbf{y};\mathbf{z})$ (y-axis). Different objectives are compared based on their target.

Concurrent work has applied the InfoMax principle in the Multi-View setting (Ji et al., 2019; Hénaff et al., 2019; Tian et al., 2019; Bachman et al., 2019), aiming to maximize mutual information between the representation $\mathbf{z}$ of a first data-view $\mathbf{x}$ and a second one $\mathbf{v}_2$. The target representation for the Multi-View InfoMax (MV-InfoMax) models should contain at least the amount of information in $\mathbf{x}$ that is predictive for $\mathbf{v}_2$, targeting the region $I(\mathbf{z};\mathbf{x}) \geq I(\mathbf{x};\mathbf{v}_2)$ on the Information Plane (purple dotted line in Figure 2). Since the MV-InfoMax has no incentive to discard any information regarding $\mathbf{x}$ from $\mathbf{z}$, a representation that is optimal according to the InfoMax principle is also optimal for any MV-InfoMax model. Our model with $\beta = 0$ (Equation 5) belongs to this family of objectives since the incentive to remove superfluous information is removed. Despite their success, Tschannen et al. (2019) has shown that the effectiveness of the InfoMax models is due to inductive biases introduced by the architecture and estimators rather than the training objective itself, since the InfoMax and MV-InfoMax objectives can be trivially maximized by using invertible encoders.

On the other hand, Variational Autoencoders (VAEs) (Kingma & Welling, 2014) define a training objective that balances compression and reconstruction error (Alemi et al., 2018) through an hyper-parameter $\beta$. Whenever $\beta$ is close to 0, the VAE objective aims for a lossless representation, approaching the same region of the Information Plane as the one targeted by InfoMax (Barber & Agakov, 2003). When $\beta$ approaches large values, the representation becomes more compressed, showing increased generalization and disentanglement (Higgins et al., 2017; Burgess et al., 2018), and, as $\beta$ approaches infinity, $I(\mathbf{z};\mathbf{x})$ goes to zero. During this transition from low to high $\beta$, however, there are no guarantees that VAEs will retain label information (Theorem B.1 in the Appendix). The path between the two regimes depends on how well the label information aligns with the induc-

tive bias introduced by encoder (Jimenez Rezende & Mohamed, 2015; Kingma et al., 2016), prior (Tomczak & Welling, 2018) and decoder architectures (Gulrajani et al., 2017; Chen et al., 2017).

The idea of discarding irrelevant information was introduced in Tishby et al. (2000) and identified as one of the possible reasons behind the generalization capabilities of deep neural networks by Tishby & Zaslavsky (2015) and Achille & Soatto (2018). Representations based on the information bottleneck principle explicitly minimize the amount of superfluous information in the representation while retaining all the label information from the data (top-left corner of the Information Plane in Figure 2). This direction of research has been explored for both single-view (Alemi et al., 2018) and multi-view settings (Wang et al., 2019), even if explicit label supervision is required to train the representation $\mathbf{z}$.

In contrast to all of the above, our work is the first to explicitly identify and discard superfluous information from the representation in the unsupervised multi-view setting. This is because unsupervised models based on the $\beta$-VAE objective remove information indiscriminately without identifying which part is relevant for teh predictive task, and the InfoMax and Multi-View InfoMax methods do not explicitly try to remove superfluous information at all. The MIB objective, on the other hand, results in the representation with the least superfluous information, i.e. the most robust among the representations that are optimal according to Multi-View InfoMax, without requiring any additional label supervision.

## 5    EXPERIMENTS

In this section we demonstrate the effectiveness of our model against state-of-the-art baselines in both the multi-view and single-view setting. In the single-view setting, we also estimate the coordinates on the Information Plane for each of the baseline methods as well as our method to validate the theory in Section 3.

The results reported in the following sections are obtained using the Jensen-Shannon $I_{JS}$ (Devon Hjelm et al., 2019; Poole et al., 2019) estimator, which resulted in better performance for MIB and the other InfoMax-based models (Table 2 in the supplementary material). In order to facilitate the comparison between the effect of the different loss functions, the same estimator is used across the different models.

### 5.1    MULTI-VIEW TASKS

We compare MIB on the sketch-based image retrieval (Sangkloy et al., 2016) and Flickr multiclass image classification (Huiskes & Lew, 2008) tasks with domain specific and prior multi-view learning methods.

#### 5.1.1    SKETCH-BASED IMAGE RETRIEVAL

**Dataset.** The Sketchy dataset (Sangkloy et al., 2016) consists of 12,500 images and 75,471 hand-drawn sketches of objects from 125 classes. As in Liu et al. (2017), we also include another 60,502 images from the ImageNet (Deng et al., 2009) from the same classes, which results in total 73,002 natural object images. As per the experimental protocol of Zhang et al. (2018), a total of 6,250 sketches (50 sketches per category) are randomly selected and removed from the training set for testing purpose, which leaves 69,221 sketches for training the model.

**Experimental Setup.** The sketch-based image retrieval task is a ranking of 73,002 natural images according to the unseen test (query) sketch. Retrieval is done for our model by generating representations for the query sketch as well as all natural images, and ranking the image by the euclidean distance of their representation from the sketch representation. The baselines use various domain specific ranking methodologies. Model performance is computed based on the class of the ranked pictures corresponding to the query sketch. The training set consists of pairs of image $\mathbf{v}_1$ and sketch $\mathbf{v}_2$ randomly selected from the same class, to ensure that both views contain the equivalent label information (mutual redundancy).

Following recent work (Zhang et al., 2018; Dutta & Akata, 2019), we use features extracted from images and sketches by a VGG (Simonyan & Zisserman, 2014) architecture trained for classification

| | | | Method | mAP@all | Prec@200 |
|---|---|---|---|---|---|
| $\mathbf{v}_1 \in \mathbb{R}^{4096}$ | $\mathbf{v}_2 \in \mathbb{R}^{4096}$ | $\mathbf{y} \in [125]$ | SaN (Yu et al., 2017) | 0.208 | 0.292 |
| | | | GN Triplet (Sangkloy et al., 2016) | 0.529 | 0.716 |
| | | "cat" | Siamese CNN (Qi et al., 2016) | 0.481 | 0.612 |
| | | | Siamese-AlexNet (Liu et al., 2017) | 0.518 | 0.690 |
| | | | Triplet-AlexNet (Liu et al., 2017) | 0.573 | 0.761 |
| | | | DSH* (Liu et al., 2017) | 0.711 | **0.866** |
| | | | GDH* (Zhang et al., 2018) | 0.810 | - |
| | | "apple" | MV-InfoMax[2] | 0.008 | 0.008 |
| | | | **MIB** | **0.856±0.005** | 0.848±0.005 |
| | | | MIB* (64-bits) | 0.851± 0.004 | 0.834±0.003 |

Table 1: Examples of the two views and class label from the Sketchy dataset (on the left) and comparison between MIB and other popular models in literature on the sketch-based image retrieval task (on the right). * denotes models that use a 64-bits binary representation. The results for MIB corresponds to $\beta = 1$.

on the TU-Berlin dataset (Eitz et al., 2012). The resulting flattened 4096-dimensional feature vectors are fed to our image and sketch encoders to produce a 64-dimensional representation. Both encoders consist of neural networks with hidden layers of 2048 and 1024 units respectively. Size of the representation and regularization strength $\beta$ are tuned on a validation sub-split. We evaluate MIB on five different train/test splits and report mean and standard deviation in Table 5.1.1. Further details on our training procedure and architecture are in Appendix G.

**Results.** Table 5.1.1 shows that the our model achieves strong performance for both mean average precision (mAP@all) and precision at 200 (Prec@200), suggesting that the representation is able to capture the common class information between the paired pictures and sketches. The effectiveness of MIB on the retrieval task can be mostly attributed to the regularization introduced with the symmetrized KL divergence between the two encoded views. In addition to discarding view-private information, this term actively aligns the representations of $\mathbf{v}_1$ and $\mathbf{v}_2$, making the MIB model especially suitable for retrieval tasks

### 5.1.2 MIR-Flickr

**Dataset.** The MIR-Flickr dataset (Huiskes & Lew, 2008) consists of 1M images annotated with 800K distinct user tags. Each image is represented by a vector of 3,857 hand-crafted image features ($\mathbf{v}_1$), while the 2,000 most frequent tags are used to produce a 2000-dimensional multi-hot encoding ($\mathbf{v}_2$) for each picture. The dataset is divided into labeled and unlabeled sets that respectively contain 975K and 25K images, where the labeled set also contains 38 distinct topic classes together with the user tags. Training images with less than two tags are removed, which reduces the total number of training samples to 749,647 pairs (Sohn et al., 2014; Wang et al., 2016). The labeled set contains 5 different splits of train, validation and test sets of size 10K/5K/10K respectively.

**Experimental Setup.** Following standard procedure in the literature (Srivastava & Salakhutdinov, 2014; Wang et al., 2016), we train our model on the unlabeled pairs of images and tags. Then a multi-label logistic classifier is trained from the representation of 10K labeled train images to the corresponding macro-categories. The quality of the representation is assessed based on the performance of the trained logistic classifier on the labeled test set. Each encoder consists of a multi-layer perceptron of 4 hidden layers with ReLU activations learning two 1024-dimensional representations $\mathbf{z}_1$ and $\mathbf{z}_2$ for images $\mathbf{v}_1$ and tags $\mathbf{v}_2$ respectively. Examples of the two views, labels, and further details on the training procedure are in Appendix G.

**Results.** Our MIB model is compared with other popular multi-view learning models in Figure 3 for $\beta = 0$ (Multi-View InfoMax), $\beta = 1$ and $\beta = 10^{-3}$ (best on validation set). Although the tuned MIB performs similarly to Multi-View InfoMax with a large number of labels, it outperforms it when fewer labels are available. Furthermore, by choosing a larger $\beta$ the accuracy of our model drastically increases in scarce label regimes, while slightly reducing the accuracy when all the labels are observed (see right side of Figure 3). This effect is likely due to a violation of the mutual

---

[2]These results are included only for completeness, as the Multi-View InfoMax objective does not produce consistent representations for the two views so there is no straight-forward way to use it for ranking.

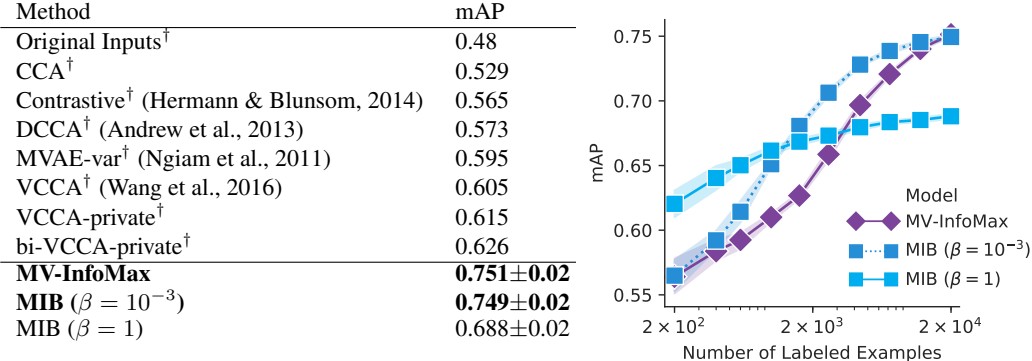

| Method | mAP |
|---|---|
| Original Inputs[†] | 0.48 |
| CCA[†] | 0.529 |
| Contrastive[†] (Hermann & Blunsom, 2014) | 0.565 |
| DCCA[†] (Andrew et al., 2013) | 0.573 |
| MVAE-var[†] (Ngiam et al., 2011) | 0.595 |
| VCCA[†] (Wang et al., 2016) | 0.605 |
| VCCA-private[†] | 0.615 |
| bi-VCCA-private[†] | 0.626 |
| **MV-InfoMax** | **0.751±0.02** |
| **MIB ($\beta = 10^{-3}$)** | **0.749±0.02** |
| MIB ($\beta = 1$) | 0.688±0.02 |

Figure 3: Left: mean average precision (mAP) of the classifier trained on different multi-view representations for the MIR-Flickr task. Right: comparing the performance for different values of $\beta$ and percentages of given labeled examples (from 1% up to 100%). Each model uses encoders of comparable size, producing a 1024 dimensional representation. [†] results from Wang et al. (2016).

redundancy constraint (see Figure 6 in the supplementary material) which can be compensated with smaller values of $\beta$ for less aggressive compression.

A possible reason for the effectiveness of MIB against some of the other baselines may be its ability to use mutual information estimators that do not require reconstruction. Both Multi-View VAE (MVAE) and Deep Variational CCA (VCCA) rely on a reconstruction term to capture cross-modal information, which can introduce bias that decreases performance.

## 5.2 Self-supervised Single-View Task

In this section, we compare the performance of different unsupervised learning models by measuring their data efficiency and empirically estimating the coordinates of their representation on the Information Plane. Since accurate estimation of mutual information is extremely expensive (McAllester & Stratos, 2018), we focus on relatively small experiments that aim to uncover the difference between popular approaches for representation learning.

**Dataset.** The dataset is generated from MNIST by creating the two views, $\mathbf{v}_1$ and $\mathbf{v}_2$, via the application of data augmentation consisting of small affine transformations and independent pixel corruption to each image. These are kept small enough to ensure that label information is not effected. Each pair of views is generated from the same underlying image, so no label information is used in this process (details in Appendix G).

**Experimental Setup.** To evaluate, we train the encoders using the unlabeled multi-view dataset just described, and then fix the representation model. A logistic regression model is trained using the resulting representations along with a subset of labels for the training set, and we report the accuracy of this model on a disjoint test set as is standard for the unsupervised representation learning literature (Tschannen et al., 2019; Tian et al., 2019; van den Oord et al., 2018). We estimate $I(\mathbf{x}; \mathbf{z})$ and $I(\mathbf{y}; \mathbf{z})$ using mutual information estimation networks trained from scratch on the final representations using batches of joint samples $\{(\boldsymbol{x}^{(i)}, \boldsymbol{y}^{(i)}, \boldsymbol{z}^{(i)})\}_{i=1}^{B} \sim p(\mathbf{x}, \mathbf{y})p_\theta(\mathbf{z}|\mathbf{x})$. All models are trained using the same encoder architecture consisting of 2 layers of 1024 hidden units with ReLU activations, resulting in 64-dimensional representations. The same data augmentation procedure was also applied for single-view architectures and models were trained for 1 million iterations with batch size $B = 64$.

**Results.** Figure 4 summarizes the results. The empirical measurements of mutual information reported on the Information Plane are consistent with the theoretical analysis reported in Section 4: models that retain less information about the data while maintaining the maximal amount of predictive information, result in better classification performance at low-label regimes, confirming the hypothesis that discarding irrelevant information yields robustness and more data-efficient represen-

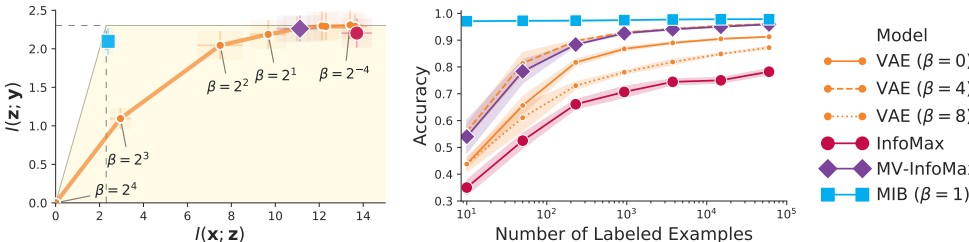

Figure 4: Comparing the representations obtained with different objectives on MNIST dataset. The empirical estimation of the coordinates on the Information Plane (in nats on the left) is followed by the respective classification accuracy for different number of randomly sampled labels (from 1 example per label up to 6000 examples per label). Representations that discard more observational information tend to perform better in scarce label regimes. The measurements used to produce the two graphs are reported in Appedix G.4.1.

tations. Notably, the MIB model with $\beta = 1$ retains almost exclusively label information, hardly decreasing the classification performance when only one label is used for each data point.

## 6 CONCLUSIONS AND FUTURE WORK

In this work, we introduce Multi-View Information Bottleneck, a novel method for taking advantage of multiple data-views to produce robust representations for downstream tasks. In our experiments, we compared MIB empirically against other approaches in the literature on three such tasks: sketch-based image retrieval, multi-view and unsupervised representation learning. The strong performance obtained in the different areas show that Multi-View Information Bottleneck can be practically applied to various tasks for which the paired observations are either readily available or artificially produced. Furthermore, the positive results on the MIR-Flickr dataset show that our model can work well in practice even when mutual redundancy holds only approximately.

There are multiple extensions that we would like to explore in future work. One interesting direction would be considering more than two views. In Appendix D we discuss why the mutual redundancy condition cannot be trivially extended to more than two views, but we still believe such an extension is possible. Secondly, we believe that exploring the role played by different choices of data augmentation could bridge the gap between the Information Bottleneck principle and the literature on invariant neural networks (Bloem-Reddy & Whye Teh, 2019), which are able to exploit known symmetries and structure of the data to remove superfluous information.

### ACKNOWLEDGMENTS

We thank Andy Keller, Karen Ullrich, Maximillian Ilse and the anonymous reviewers for their feedback and insightful comments. This work has received funding from the ERC under the Horizon 2020 program (grant agreement No. 853489). The Titan Xp and Titan V used for this research were donated by the NVIDIA Corporation.

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

## A  PROPERTIES OF MUTUAL INFORMATION AND ENTROPY

In this section we enumerate some of the properties of mutual information that are used to prove the theorems reported in this work. For any random variables $\mathbf{w}$, $\mathbf{x}$, $\mathbf{y}$ and $\mathbf{z}$:

$(P_1)$  Positivity:

$$I(\mathbf{x};\mathbf{y}) \geq 0, I(\mathbf{x};\mathbf{y}|\mathbf{z}) \geq 0$$

$(P_2)$  Chain rule:

$$I(\mathbf{xy};\mathbf{z}) = I(\mathbf{y};\mathbf{z}) + I(\mathbf{x};\mathbf{z}|\mathbf{y})$$

$(P_3)$  Chain rule (Multivariate Mutual Information):

$$I(\mathbf{x};\mathbf{y};\mathbf{z}) = I(\mathbf{y};\mathbf{z}) - I(\mathbf{y};\mathbf{z}|\mathbf{x})$$

$(P_4)$  Positivity of discrete entropy:
For discrete $\mathbf{x}$

$$H(\mathbf{x}) \geq 0, H(\mathbf{x}|\mathbf{y}) \geq 0$$

$(P_5)$  Entropy and Mutual Information

$$H(\mathbf{x}) = H(\mathbf{x}|\mathbf{y}) + I(\mathbf{x};\mathbf{y})$$

## B  THEOREMS AND PROOFS

In the following section we prove the statements reported in the main text of the paper. Whenever a random variable $\mathbf{z}$ is defined to be a representation of another random variable $\mathbf{x}$, we state that $\mathbf{z}$ is conditionally independent from any other variable in the system once $\mathbf{x}$ is observed. This does not imply that $\mathbf{z}$ must be a deterministic function of $\mathbf{x}$, but that the source of stochasticity for $\mathbf{z}$ is independent of the other random variables. As a result whenever $\mathbf{z}$ is a representation of $\mathbf{x}$:

$$I(\mathbf{z};\mathbf{a}|\mathbf{xb}) = 0,$$

for any variable (or groups of variables) $\mathbf{a}$ and $\mathbf{b}$ in the system.

### B.1  ON SUFFICIENCY

**Proposition B.1.** *Let $\mathbf{x}$ and $\mathbf{y}$ be random variables with joint distribution $p(\mathbf{x},\mathbf{y})$. Let $\mathbf{z}$ be a representation of $\mathbf{x}$, then $\mathbf{z}$ is sufficient for $\mathbf{y}$ if and only if $I(\mathbf{x};\mathbf{y}) = I(\mathbf{y};\mathbf{z})$*

*Hypothesis:*

$(H_1)$  $\mathbf{z}$ *is a representation of* $\mathbf{x}$*:* $I(\mathbf{y};\mathbf{z}|\mathbf{x}) = 0$

*Thesis:*

$(T_1)$  $I(\mathbf{x};\mathbf{y}|\mathbf{z}) = 0 \iff I(\mathbf{x};\mathbf{y}) = I(\mathbf{y};\mathbf{z})$

*Proof.*

$$I(\mathbf{x};\mathbf{y}|\mathbf{z}) \overset{(P_3)}{=} I(\mathbf{x};\mathbf{y}) - I(\mathbf{x};\mathbf{y};\mathbf{z}) \overset{(P_3)}{=} I(\mathbf{x};\mathbf{y}) - I(\mathbf{y};\mathbf{z}) - I(\mathbf{y};\mathbf{z}|\mathbf{x})$$
$$\overset{(H_1)}{=} I(\mathbf{x};\mathbf{y}) - I(\mathbf{y};\mathbf{z})$$

Since both $I(\mathbf{x};\mathbf{y})$ and $I(\mathbf{y};\mathbf{z})$ are non-negative $(P_1)$, $I(\mathbf{x};\mathbf{y}|\mathbf{z}) = 0 \iff I(\mathbf{y};\mathbf{z}) = I(\mathbf{x};\mathbf{y})$  $\square$

## B.2 No Free Generalization

**Theorem B.1.** *Let* $\mathbf{x}$*,* $\mathbf{z}$ *and* $\mathbf{y}$ *be random variables with joint distribution* $p(\mathbf{x}, \mathbf{y}, \mathbf{z})$*. Let* $\mathbf{z}'$ *be a representation of* $\mathbf{x}$ *that satisfies* $I(\mathbf{x}; \mathbf{z}) > I(\mathbf{x}; \mathbf{z}')$*, then it is always possible to find a label* $\mathbf{y}$ *for which* $\mathbf{z}'$ *is not predictive for* $\mathbf{y}$ *while* $\mathbf{z}$ *is.*

*Hypothesis:*

$(H_1)$  $\mathbf{z}'$ *is a representation of* $\mathbf{x}$*:* $I(\mathbf{y}; \mathbf{z}'|\mathbf{x}) = 0$

$(H_2)$  $I(\mathbf{x}; \mathbf{z}) > I(\mathbf{x}; \mathbf{z}')$

*Thesis:*

$(T_1)$  $I(\mathbf{x}; \mathbf{z}') < I(\mathbf{x}; \mathbf{z}) \implies \exists \mathbf{y}. I(\mathbf{y}; \mathbf{z}) > I(\mathbf{y}; \mathbf{z}') = 0$

*Proof.* By construction.

1. We first factorize $\mathbf{x}$ as a function of two independent random variables (Proposition 2.1 Achille & Soatto (2018)) by picking $\mathbf{y}$ such that:

   $(C_1)$  $I(\mathbf{y}; \mathbf{z}') = 0$
   $(C_2)$  $\mathbf{x} = f(\mathbf{z}', \mathbf{y})$

   for some deterministic function $f$. Note that such $\mathbf{y}$ always exists.

2. Since $\mathbf{x}$ is a function of $\mathbf{y}$ and $\mathbf{z}'$:

   $(C_4)$  $I(\mathbf{x}; \mathbf{z}|\mathbf{y}\mathbf{z}') = 0$

Considering $I(\mathbf{y}; \mathbf{z})$:

$$
\begin{aligned}
I(\mathbf{y}; \mathbf{z}) &\overset{(P_3)}{=} I(\mathbf{y}; \mathbf{z}|\mathbf{x}) + I(\mathbf{x}; \mathbf{y}; \mathbf{z}) \\
&\overset{(P_1)}{\geq} I(\mathbf{x}; \mathbf{y}; \mathbf{z}) \\
&\overset{(P_3)}{=} I(\mathbf{x}; \mathbf{z}) - I(\mathbf{x}; \mathbf{z}|\mathbf{y}) \\
&\overset{(P_3)}{=} I(\mathbf{x}; \mathbf{z}) - I(\mathbf{x}; \mathbf{z}|\mathbf{y}\mathbf{z}') - I(\mathbf{x}; \mathbf{z}; \mathbf{z}'|\mathbf{y}) \\
&\overset{(C_2)}{=} I(\mathbf{x}; \mathbf{z}) - I(\mathbf{x}; \mathbf{z}; \mathbf{z}'|\mathbf{y}) \\
&\overset{(P_3)}{=} I(\mathbf{x}; \mathbf{z}) - I(\mathbf{x}; \mathbf{z}'|\mathbf{y}) + I(\mathbf{x}; \mathbf{z}'|\mathbf{y}\mathbf{z}) \\
&\overset{(P_1)}{\geq} I(\mathbf{x}; \mathbf{z}) - I(\mathbf{x}; \mathbf{z}'|\mathbf{y}) \\
&\overset{(P_3)}{=} I(\mathbf{x}; \mathbf{z}) - I(\mathbf{x}; \mathbf{z}') + I(\mathbf{x}; \mathbf{y}; \mathbf{z}') \\
&\overset{(P_3)}{=} I(\mathbf{x}; \mathbf{z}) - I(\mathbf{x}; \mathbf{z}') + I(\mathbf{y}; \mathbf{z}') - I(\mathbf{y}; \mathbf{z}'|\mathbf{x}) \\
&\overset{(P_1)}{\geq} I(\mathbf{x}; \mathbf{z}) - I(\mathbf{x}; \mathbf{z}') - I(\mathbf{y}; \mathbf{z}'|\mathbf{x}) \\
&\overset{(H_1)}{=} I(\mathbf{x}; \mathbf{z}) - I(\mathbf{x}; \mathbf{z}') \\
&\overset{(H_2)}{>} 0
\end{aligned}
$$

Since $I(\mathbf{y}; \mathbf{z}') = 0$ by construction and $I(\mathbf{y}; \mathbf{z}) > 0$, the $\mathbf{y}$ built in 1. satisfies the conditions reported in the thesis. $\qquad\square$

**Corollary B.1.1.** *Let $\mathbf{z}'$ be a representation of $\mathbf{x}$ that discards observational information. There is always a label $\mathbf{y}$ for which a $\mathbf{z}'$ is not predictive, while the original observations are.*
*Hypothesis:*

$(H_1)$ $\mathbf{x}$ *is discrete*

$(H_2)$ $\mathbf{z}'$ *discards information regarding $\mathbf{x}$:* $I(\mathbf{z}'; \mathbf{x}) < H(\mathbf{x})$

*Thesis:*

$(T_1)$ $\exists \mathbf{y}. I(\mathbf{y}; \mathbf{x}) > I(\mathbf{y}; \mathbf{z}') = 0$

*Proof.* By construction using Theorem B.1.

1. Set $\mathbf{z} = \mathbf{x}$:

$(C_1)$ $I(\mathbf{x}; \mathbf{z}) \overset{(P_5)}{=} H(\mathbf{x}) - H(\mathbf{x}|\mathbf{z}) \overset{(H_1)}{=} H(\mathbf{x})$

2. $I(\mathbf{z}'; \mathbf{x}) < H(\mathbf{x}) \overset{(C_1)}{\Longrightarrow} I(\mathbf{z}'; \mathbf{x}) < I(\mathbf{x}; \mathbf{z})$

Since the hypothesis are met, we conclude that there exist $\mathbf{y}$ such that $I(\mathbf{y}; \mathbf{x}) > I(\mathbf{y}; \mathbf{z}') = 0$  □

## B.3 MULTI-VIEW

### B.3.1 MULTI-VIEW REDUNDANCY AND SUFFICIENCY

**Proposition B.2.** *Let $\mathbf{v}_1$, $\mathbf{v}_2$, $\mathbf{y}$ be random variables with joint distribution $p(\mathbf{v}_1, \mathbf{v}_2, \mathbf{y})$. Let $\mathbf{z}_1$ be a representation of $\mathbf{v}_1$, then:*

$$I(\mathbf{v}_1; \mathbf{y}|\mathbf{z}_1) \leq I(\mathbf{v}_1; \mathbf{v}_2|\mathbf{z}_1) + I(\mathbf{v}_1; \mathbf{y}|\mathbf{v}_2)$$

*Hypothesis:*

$(H_1)$ $\mathbf{z}_1$ *is a representation of $\mathbf{v}_1$:* $I(\mathbf{y}; \mathbf{z}_1|\mathbf{v}_2\mathbf{v}_1) = 0$

*Thesis:*

$(T_1)$ $I(\mathbf{v}_1; \mathbf{y}|\mathbf{z}_1) \leq I(\mathbf{v}_1; \mathbf{v}_2|\mathbf{z}_1) + I(\mathbf{v}_1; \mathbf{y}|\mathbf{v}_2)$

*Proof.* Since $\mathbf{z}_1$ is a representation of $\mathbf{v}_1$:

$(C_1)$ $I(\mathbf{y}; \mathbf{z}_1|\mathbf{v}_2\mathbf{v}_1) = 0$

Therefore:

$$
\begin{aligned}
I(\mathbf{v}_1; \mathbf{y}|\mathbf{z}_1) &\overset{(P_3)}{=} I(\mathbf{v}_1; \mathbf{y}|\mathbf{z}_1\mathbf{v}_2) + I(\mathbf{v}_1; \mathbf{v}_2; \mathbf{y}|\mathbf{z}_1) \\
&\overset{(P_3)}{=} I(\mathbf{v}_1; \mathbf{y}|\mathbf{v}_2) - I(\mathbf{v}_1; \mathbf{y}; \mathbf{z}_1|\mathbf{v}_2) + I(\mathbf{v}_1; \mathbf{v}_2; \mathbf{y}|\mathbf{z}_1) \\
&\overset{(P_3)}{=} I(\mathbf{v}_1; \mathbf{y}|\mathbf{v}_2) - I(\mathbf{y}; \mathbf{z}_1|\mathbf{v}_2) + I(\mathbf{y}; \mathbf{z}_1|\mathbf{v}_2\mathbf{v}_1) + I(\mathbf{v}_1; \mathbf{v}_2; \mathbf{y}|\mathbf{z}_1) \\
&\overset{(P_1)}{\leq} I(\mathbf{v}_1; \mathbf{y}|\mathbf{v}_2) + I(\mathbf{y}; \mathbf{z}_1|\mathbf{v}_2\mathbf{v}_1) + I(\mathbf{v}_1; \mathbf{v}_2; \mathbf{y}|\mathbf{z}_1) \\
&\overset{(H_1)}{=} I(\mathbf{v}_1; \mathbf{y}|\mathbf{v}_2) + I(\mathbf{v}_1; \mathbf{v}_2; \mathbf{y}|\mathbf{z}_1) \\
&\overset{(P_3)}{=} I(\mathbf{v}_1; \mathbf{y}|\mathbf{v}_2) + I(\mathbf{v}_1; \mathbf{v}_2|\mathbf{z}_1) - I(\mathbf{v}_1; \mathbf{v}_2|\mathbf{z}_1\mathbf{y}) \\
&\overset{(P_1)}{\leq} I(\mathbf{v}_1; \mathbf{y}|\mathbf{v}_2) + I(\mathbf{v}_1; \mathbf{v}_2|\mathbf{z}_1)
\end{aligned}
$$

□

**Proposition B.3.** *Let $\mathbf{v}_1$ be a redundant view with respect to $\mathbf{v}_2$ for $\mathbf{y}$. Any representation $\mathbf{z}_1$ of $\mathbf{v}_1$ that is sufficient for $\mathbf{v}_2$ is also sufficient for $\mathbf{y}$.*

*Hypothesis:*

($H_1$) $\mathbf{z}_1$ *is a representation of* $\mathbf{v}_1$: $I(\mathbf{y}; \mathbf{z}_1 | \mathbf{v}_2 \mathbf{v}_1) = 0$

($H_2$) $\mathbf{v}_1$ *is redundant with respect to* $\mathbf{v}_2$ *for* $\mathbf{y}$: $I(\mathbf{y}; \mathbf{v}_1 | \mathbf{v}_2) = 0$

*Thesis:*

($T_1$) $I(\mathbf{v}_1; \mathbf{v}_2 | \mathbf{z}_1) = 0 \implies I(\mathbf{v}_1; \mathbf{y} | \mathbf{z}_1) = 0$

*Proof.* Using the results from Theorem B.2:

$$I(\mathbf{v}_1; \mathbf{y} | \mathbf{z}_1) \overset{(Th_{B.2})}{\leq} I(\mathbf{v}_1; \mathbf{y} | \mathbf{v}_2) + I(\mathbf{v}_1; \mathbf{v}_2 | \mathbf{z}_1) \overset{(H_2)}{=} I(\mathbf{v}_1; \mathbf{v}_2 | \mathbf{z}_1)$$

Therefore $I(\mathbf{v}_1; \mathbf{v}_2 | \mathbf{z}_1) = 0 \implies I(\mathbf{v}_1; \mathbf{y} | \mathbf{z}_1) = 0$ $\qquad\qquad\qquad\qquad\qquad\qquad$ $\square$

**Theorem B.2.** *Let $\mathbf{v}_1$, $\mathbf{v}_2$ and $\mathbf{y}$ be random variables with distribution $p(\mathbf{v}_1, \mathbf{v}_2, \mathbf{y})$. Let $\mathbf{z}$ be a representation of $\mathbf{v}_1$, then*

$$I(\mathbf{y}; \mathbf{z}_1) \geq I(\mathbf{y}; \mathbf{v}_1 \mathbf{v}_2) - I(\mathbf{v}_1; \mathbf{v}_2 | \mathbf{z}_1) - I(\mathbf{v}_1; \mathbf{y} | \mathbf{v}_2) - I(\mathbf{v}_2; \mathbf{y} | \mathbf{v}_1)$$

*Hypothesis:*

($H_1$) $\mathbf{z}_1$ *is a representation of* $\mathbf{v}_1$: $I(\mathbf{y}; \mathbf{z}_1 | \mathbf{v}_1 \mathbf{v}_2) = 0$

*Thesis:*

($T_1$) $I(\mathbf{y}; \mathbf{z}_1) \geq I(\mathbf{y}; \mathbf{v}_1 \mathbf{v}_2) - I(\mathbf{v}_1; \mathbf{v}_2 | \mathbf{z}_1) - I(\mathbf{v}_1; \mathbf{y} | \mathbf{v}_2) - I(\mathbf{v}_2; \mathbf{y} | \mathbf{v}_1)$

*Proof.*

$$
\begin{aligned}
I(\mathbf{y}; \mathbf{z}_1) \overset{(P_3)}{=} & \; I(\mathbf{y}; \mathbf{z}_1 | \mathbf{v}_1 \mathbf{v}_2) + I(\mathbf{y}; \mathbf{v}_1 \mathbf{v}_2; \mathbf{z}_1) \\
\overset{(H_1)}{=} & \; I(\mathbf{y}; \mathbf{v}_1 \mathbf{v}_2; \mathbf{z}_1) \\
\overset{(P_3)}{=} & \; I(\mathbf{y}; \mathbf{v}_1 \mathbf{v}_2) - I(\mathbf{y}; \mathbf{v}_1 \mathbf{v}_2 | \mathbf{z}_1) \\
\overset{(P_2)}{=} & \; I(\mathbf{y}; \mathbf{v}_1 \mathbf{v}_2) - I(\mathbf{y}; \mathbf{v}_1 | \mathbf{z}_1) - I(\mathbf{y}; \mathbf{v}_2 | \mathbf{z}_1 \mathbf{v}_1) \\
\overset{(P_3)}{=} & \; I(\mathbf{y}; \mathbf{v}_1 \mathbf{v}_2) - I(\mathbf{y}; \mathbf{v}_1 | \mathbf{z}_1) - I(\mathbf{y}; \mathbf{v}_2 | \mathbf{v}_1) + I(\mathbf{y}; \mathbf{v}_2; \mathbf{z}_1 | \mathbf{v}_1) \\
\overset{(P_3)}{=} & \; I(\mathbf{y}; \mathbf{v}_1 \mathbf{v}_2) - I(\mathbf{y}; \mathbf{v}_1 | \mathbf{z}_1) - I(\mathbf{y}; \mathbf{v}_2 | \mathbf{v}_1) + I(\mathbf{y}; \mathbf{z}_1 | \mathbf{v}_1) - I(\mathbf{y}; \mathbf{z}_1 | \mathbf{v}_1 \mathbf{v}_2) \\
\overset{(H_1)}{=} & \; I(\mathbf{y}; \mathbf{v}_1 \mathbf{v}_2) - I(\mathbf{y}; \mathbf{v}_1 | \mathbf{z}_1) - I(\mathbf{y}; \mathbf{v}_2 | \mathbf{v}_1) + I(\mathbf{y}; \mathbf{z}_1 | \mathbf{v}_1) \\
\overset{(P_1)}{\geq} & \; I(\mathbf{y}; \mathbf{v}_1 \mathbf{v}_2) - I(\mathbf{y}; \mathbf{v}_1 | \mathbf{z}_1) - I(\mathbf{y}; \mathbf{v}_2 | \mathbf{v}_1) \\
\overset{(Prop_{B.2})}{\geq} & \; I(\mathbf{y}; \mathbf{v}_1 \mathbf{v}_2) - I(\mathbf{v}_1; \mathbf{y} | \mathbf{v}_2) - I(\mathbf{v}_1; \mathbf{v}_2 | \mathbf{z}_1) - I(\mathbf{y}; \mathbf{v}_2 | \mathbf{v}_1)
\end{aligned}
$$

$\qquad\qquad\qquad\qquad\qquad\qquad\qquad\qquad\qquad\qquad\qquad\qquad\qquad\qquad\qquad$ $\square$

**Corollary B.2.1.** *Let $\mathbf{v}_1$ and $\mathbf{v}_2$ be mutually redundant views for $\mathbf{y}$. Let $\mathbf{z}_1$ be a representation of $\mathbf{v}_1$ that is sufficient for $\mathbf{v}_2$. Then:*

$$I(\mathbf{y}; \mathbf{z}_1) = I(\mathbf{v}_1 \mathbf{v}_2; \mathbf{y})$$

*Hypothesis:*

($H_1$) $\mathbf{z}_1$ *is a representation of* $\mathbf{v}_1$: $I(\mathbf{y}; \mathbf{z}_1 | \mathbf{v}_1 \mathbf{v}_2) = 0$

($H_2$) $\mathbf{v}_1$ *and* $\mathbf{v}_2$ *are mutually redundant for* $\mathbf{y}$: $I(\mathbf{y}; \mathbf{v}_1 | \mathbf{v}_2) + I(\mathbf{y}; \mathbf{v}_2 | \mathbf{v}_1) = 0$

($H_3$) $\mathbf{z}_1$ *is sufficient for* $\mathbf{v}_2$: $I(\mathbf{v}_2; \mathbf{v}_1 | \mathbf{z}) = 0$

*Thesis:*

($T_1$) $I(\mathbf{y}; \mathbf{z}_1) = I(\mathbf{v}_1 \mathbf{v}_2; \mathbf{y})$

*Proof.* Using Theorem B.2

$$
\begin{aligned}
I(\mathbf{y}; \mathbf{z}_1) &\overset{(Th_{B.2})}{\geq} I(\mathbf{y}; \mathbf{v}_1 \mathbf{v}_2) - I(\mathbf{v}_1; \mathbf{y} | \mathbf{v}_2) - I(\mathbf{v}_1; \mathbf{v}_2 | \mathbf{z}_1) - I(\mathbf{y}; \mathbf{v}_2 | \mathbf{v}_1) \\
&\overset{(H_2)}{=} I(\mathbf{y}; \mathbf{v}_1 \mathbf{v}_2) - I(\mathbf{v}_1; \mathbf{v}_2 | \mathbf{z}_1) \\
&\overset{(H_3)}{=} I(\mathbf{y}; \mathbf{v}_1 \mathbf{v}_2)
\end{aligned}
$$

Since $I(\mathbf{y}; \mathbf{z}_1) \leq I(\mathbf{y}; \mathbf{v}_1 \mathbf{v}_2)$ is a consequence of the data processing inequality, we conclude that $I(\mathbf{y}; \mathbf{z}_1) = I(\mathbf{y}; \mathbf{v}_1 \mathbf{v}_2)$ □

### B.4 SUFFICIENCY AND AUGMENTATION

Let $\mathbf{x}$ and $\mathbf{y}$ be random variables with domain $\mathbb{X}$ and $\mathbb{Y}$ respectively. Let $\mathbb{T}$ be a class of functions $t : \mathbb{X} \to \mathbb{W}$ and let $\mathbf{t}_1$ and $\mathbf{t}_2$ be a random variables over $\mathbb{T}$ that depends only on $\mathbf{x}$. For the theorems and corollaries discussed in this section, we are going to consider the independence assumption that can be derived from the graphical model $\mathcal{G}$ reported in Figure 5.

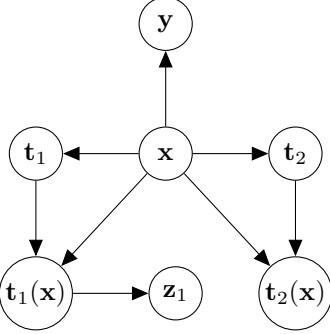

Figure 5: Visualization of the graphical model $\mathcal{G}$ that relates the observations $\mathbf{x}$, label $\mathbf{y}$, functions used for augmentation $\mathbf{t}_1$, $\mathbf{t}_2$ and the representation $\mathbf{z}_1$.

**Proposition B.4.** *Whenever* $I(\mathbf{t}_1(\mathbf{x}); \mathbf{y}) = I(\mathbf{t}_2(\mathbf{x}); \mathbf{y}) = I(\mathbf{x}; \mathbf{y})$ *the two views* $\mathbf{t}_1(\mathbf{x})$ *and* $\mathbf{t}_2(\mathbf{x})$ *must be mutually redundant for* $\mathbf{y}$.

*Hypothesis:*

($H_1$) *Independence relations determined by* $\mathcal{G}$

*Thesis:*

($T_1$) $I(\mathbf{t}_1(\mathbf{x}); \mathbf{y}) = I(\mathbf{t}_2(\mathbf{x}); \mathbf{y}) = I(\mathbf{x}; \mathbf{y}) \implies I(\mathbf{t}_1(\mathbf{x}); \mathbf{y} | \mathbf{t}_2(\mathbf{x})) + I(\mathbf{t}_2(\mathbf{x}); \mathbf{y} | \mathbf{t}_1(\mathbf{x})) = 0$

*Proof.*

1. Considering $\mathcal{G}$ we have:

$(C_1)$ $I(\mathbf{t}_1(\mathbf{x}); \mathbf{y}|\mathbf{x}\mathbf{t}_2(\mathbf{x})) = 0$

$(C_2)$ $I(\mathbf{y}; \mathbf{t}_2(\mathbf{x})|\mathbf{x}) = 0$

2. Since $\mathbf{t}_2(\mathbf{x})$ is uniquely determined by $\mathbf{x}$ and $\mathbf{t}_2$:

$(C_3)$ $I(\mathbf{t}_2(\mathbf{x}); \mathbf{y}|\mathbf{x}\mathbf{t}_2) = 0$

3. Consider $I(\mathbf{t}_1(\mathbf{x}); \mathbf{y}|\mathbf{t}_2(\mathbf{x}))$

$$
\begin{aligned}
I(\mathbf{t}_1(\mathbf{x}); \mathbf{y}|\mathbf{t}_2(\mathbf{x})) &\overset{(P_3)}{=} I(\mathbf{t}_1(\mathbf{x}); \mathbf{y}|\mathbf{x}\mathbf{t}_2(\mathbf{x})) + I(\mathbf{t}_1(\mathbf{x}); \mathbf{y}; \mathbf{x}|\mathbf{t}_2(\mathbf{x})) \\
&\overset{(C_1)}{=} I(\mathbf{t}_1(\mathbf{x}); \mathbf{y}; \mathbf{x}|\mathbf{t}_2(\mathbf{x})) \\
&\overset{(P_3)}{=} I(\mathbf{y}; \mathbf{x}|\mathbf{t}_2(\mathbf{x})) - I(\mathbf{y}; \mathbf{x}|\mathbf{t}_1(\mathbf{x})\mathbf{t}_2(\mathbf{x})) \\
&\overset{(P_1)}{\leq} I(\mathbf{y}; \mathbf{x}|\mathbf{t}_2(\mathbf{x})) \\
&\overset{(P_3)}{=} I(\mathbf{y}; \mathbf{x}) - I(\mathbf{y}; \mathbf{x}; \mathbf{t}_2(\mathbf{x})) \\
&\overset{(P_3)}{=} I(\mathbf{y}; \mathbf{x}) - I(\mathbf{y}; \mathbf{t}_2(\mathbf{x})) + I(\mathbf{y}; \mathbf{t}_2(\mathbf{x})|\mathbf{x}) \\
&\overset{(P_3)}{=} I(\mathbf{y}; \mathbf{x}) - I(\mathbf{y}; \mathbf{t}_2(\mathbf{x})) + I(\mathbf{y}; \mathbf{t}_2(\mathbf{x})|\mathbf{t}_2\mathbf{x}) + I(\mathbf{y}; \mathbf{t}_2(\mathbf{x}); \mathbf{t}_2|\mathbf{x}) \\
&\overset{(C_3)}{=} I(\mathbf{y}; \mathbf{x}) - I(\mathbf{y}; \mathbf{t}_2(\mathbf{x})) + I(\mathbf{y}; \mathbf{t}_2(\mathbf{x}); \mathbf{t}_2|\mathbf{x}) \\
&\overset{(P_3)}{=} I(\mathbf{y}; \mathbf{x}) - I(\mathbf{y}; \mathbf{t}_2(\mathbf{x})) + I(\mathbf{y}; \mathbf{t}_2(\mathbf{x})|\mathbf{x}) - I(\mathbf{y}; \mathbf{t}_2(\mathbf{x})|\mathbf{t}_2\mathbf{x}) \\
&\overset{(P_1)}{\geq} I(\mathbf{y}; \mathbf{x}) - I(\mathbf{y}; \mathbf{t}_2(\mathbf{x})) + I(\mathbf{y}; \mathbf{t}_2(\mathbf{x})|\mathbf{x}) \\
&\overset{(C_2)}{\geq} I(\mathbf{y}; \mathbf{x}) - I(\mathbf{y}; \mathbf{t}_2(\mathbf{x}))
\end{aligned}
$$

Therefore $I(\mathbf{y}; \mathbf{x}) = I(\mathbf{y}; \mathbf{t}_2(\mathbf{x})) \implies I(\mathbf{t}_1(\mathbf{x}); \mathbf{y}|\mathbf{t}_2(\mathbf{x})) = 0$

The proof for $I(\mathbf{y}; \mathbf{x}) = I(\mathbf{y}; \mathbf{t}_1(\mathbf{x})) \implies I(\mathbf{t}_2(\mathbf{x}); \mathbf{y}|\mathbf{t}_1(\mathbf{x})) = 0$ is symmetric, therefore we conclude $I(\mathbf{t}_1(\mathbf{x}); \mathbf{y}) = I(\mathbf{t}_2(\mathbf{x}); \mathbf{y}) = I(\mathbf{x}; \mathbf{y}) \implies I(\mathbf{t}_1(\mathbf{x}); \mathbf{y}|\mathbf{t}_2(\mathbf{x})) + I(\mathbf{t}_2(\mathbf{x}); \mathbf{y}|\mathbf{t}_1(\mathbf{x})) = 0$ $\square$

**Theorem B.3.** *Let* $I(\mathbf{t}_1(\mathbf{x}); \mathbf{y}) = I(\mathbf{t}_2(\mathbf{x}); \mathbf{y}) = I(\mathbf{x}; \mathbf{y})$. *Let* $\mathbf{z}_1$ *be a representation of* $\mathbf{t}_1(\mathbf{x})$ . *If* $\mathbf{z}_1$ *is sufficient for* $\mathbf{t}_2(\mathbf{x})$ *then* $I(\mathbf{x}; \mathbf{y}) = I(\mathbf{y}; \mathbf{z}_1)$.

*Hypothesis:*

$(H_1)$ *Independence relations determined by* $\mathcal{G}$

$(H_2)$ $I(\mathbf{t}_1(\mathbf{x}); \mathbf{y}) = I(\mathbf{t}_2(\mathbf{x}); \mathbf{y}) = I(\mathbf{x}; \mathbf{y})$

*Thesis:*

$(T_1)$ $I(\mathbf{t}_1(\mathbf{x}); \mathbf{t}_2(\mathbf{x})|\mathbf{z}_1) = 0 \implies I(\mathbf{x}; \mathbf{y}) = I(\mathbf{y}; \mathbf{z}_1)$

*Proof.* Since $\mathbf{t}_1(\mathbf{x})$ is redundant for $\mathbf{t}_2(\mathbf{x})$ (Proposition B.4) any representation $\mathbf{z}_1$ of $\mathbf{t}_1(\mathbf{x})$ that is sufficient for $\mathbf{t}_2(\mathbf{x})$ must also be sufficient for $\mathbf{y}$ (Theorem B.2). Using Proposition B.1 we have $I(\mathbf{y}; \mathbf{z}_1) = I(\mathbf{y}; \mathbf{t}_1(\mathbf{x}))$. Since $I(\mathbf{y}; \mathbf{t}_1(\mathbf{x})) = I(\mathbf{y}; \mathbf{x})$ by hypothesis, we conclude $I(\mathbf{x}; \mathbf{y}) = I(\mathbf{y}; \mathbf{z}_1)$ $\square$

## C  INFORMATION PLANE

Every representation $\mathbf{z}$ of $\mathbf{x}$ must satisfy the following constraints:

- $0 \leq I(\mathbf{y}; \mathbf{z}) \leq I(\mathbf{x}; \mathbf{y})$: The amount of label information ranges from 0 to the total predictive information accessible from the raw observations $I(\mathbf{x}; \mathbf{y})$.

- $I(\mathbf{y}; \mathbf{z}) \leq I(\mathbf{x}; \mathbf{z}) \leq I(\mathbf{y}; \mathbf{z}) + H(\mathbf{x}|\mathbf{y})$: The representation must contain more information about the observations than about the label. When $\mathbf{x}$ is discrete, the amount of discarded label information $I(\mathbf{x}; \mathbf{y}) - I(\mathbf{y}; \mathbf{z})$ must be smaller than the amount of discarded observational information $H(\mathbf{x}) - I(\mathbf{x}; \mathbf{z})$, which implies $I(\mathbf{x}; \mathbf{z}) \leq I(\mathbf{y}; \mathbf{z}) + H(\mathbf{x}|\mathbf{y})$.

*Proof.* Since $\mathbf{z}$ is a representation of $\mathbf{x}$:

$(C_1)$ $I(\mathbf{y}; \mathbf{z}|\mathbf{x}) = 0$

Considering the four bounds separately:

1. $I(\mathbf{y}; \mathbf{z}) \geq 0$: Follows from $P_1$

2. $I(\mathbf{x}; \mathbf{z}) \geq I(\mathbf{y}; \mathbf{z})$: Follows from:

$$
\begin{aligned}
I(\mathbf{x}; \mathbf{z}) &\stackrel{(P_3)}{=} I(\mathbf{x}; \mathbf{z}|\mathbf{y}) + I(\mathbf{x}; \mathbf{y}; \mathbf{z}) \\
&\stackrel{(P_1)}{\geq} I(\mathbf{x}; \mathbf{y}; \mathbf{z}) \\
&\stackrel{(P_3)}{=} I(\mathbf{y}; \mathbf{z}) - I(\mathbf{y}; \mathbf{z}|\mathbf{x}) \\
&\stackrel{(C_1)}{=} I(\mathbf{y}; \mathbf{z})
\end{aligned}
$$

3. $I(\mathbf{y}; \mathbf{z}) \leq I(\mathbf{y}; \mathbf{x})$: Data processing inequality

$$
\begin{aligned}
I(\mathbf{y}; \mathbf{z}) &\stackrel{(P_3)}{=} I(\mathbf{y}; \mathbf{z}|\mathbf{x}) + I(\mathbf{y}; \mathbf{z}; \mathbf{x}) \\
&\stackrel{(C_1)}{=} I(\mathbf{y}; \mathbf{z}; \mathbf{x}) \\
&\stackrel{(P_3)}{=} I(\mathbf{x}; \mathbf{y}) - I(\mathbf{x}; \mathbf{y}|\mathbf{z}) \\
&\stackrel{(P_1)}{\leq} I(\mathbf{x}; \mathbf{y})
\end{aligned}
$$

4. $I(\mathbf{x}; \mathbf{z}) \leq I(\mathbf{y}; \mathbf{z}) + H(\mathbf{x}|\mathbf{y})$: For discrete $\mathbf{x}$:

$$
\begin{aligned}
I(\mathbf{x}; \mathbf{z}) &\stackrel{(P_3)}{=} I(\mathbf{x}; \mathbf{z}|\mathbf{y}) + I(\mathbf{x}; \mathbf{y}; \mathbf{z}) \\
&\stackrel{(P_3)}{=} I(\mathbf{x}; \mathbf{z}|\mathbf{y}) + I(\mathbf{y}; \mathbf{z}) - I(\mathbf{y}; \mathbf{z}|\mathbf{x}) \\
&\stackrel{(C_1)}{=} I(\mathbf{x}; \mathbf{z}|\mathbf{y}) + I(\mathbf{y}; \mathbf{z}) \\
&\stackrel{(P_4)}{\leq} I(\mathbf{x}; \mathbf{z}|\mathbf{y}) + H(\mathbf{x}|\mathbf{y}\mathbf{z}) + I(\mathbf{y}; \mathbf{z}) \\
&\stackrel{(P_5)}{=} H(\mathbf{x}|\mathbf{y}) + I(\mathbf{y}; \mathbf{z})
\end{aligned}
$$

$\square$

Note that the discreetness of $\mathbf{x}$ is required only to prove bound 4. For continuous $\mathbf{x}$ bounds 1, 2 and 3 still hold.

## D  NON-TRANSITIVITY OF MUTUAL REDUNDANCY

The mutual redundancy condition between two views $\mathbf{v}_1$ and $\mathbf{v}_2$ for a label $\mathbf{y}$ can not be trivially extended to an arbitrary number of views, as the relation is not transitive because of some higher order interaction between the different views and the label. This can be shown with a simple example.

Given three views $\mathbf{v}_1$, $\mathbf{v}_2$ and $\mathbf{v}_3$ and a task $\mathbf{y}$ such that:

- $\mathbf{v}_1$ and $\mathbf{v}_2$ are mutually redundant for $\mathbf{y}$
- $\mathbf{v}_2$ and $\mathbf{v}_3$ are mutually redundant for $\mathbf{y}$

Then, we show that $\mathbf{v}_1$ is not necessarily mutually redundant with respect to $\mathbf{v}_3$ for $\mathbf{y}$.

Let $\mathbf{v}_1$, $\mathbf{v}_2$ and $\mathbf{v}_3$ be fair and independent binary random variables. Defining $\mathbf{y}$ as the exclusive or operator applied to $\mathbf{v}_1$ and $\mathbf{v}_3$ ( $\mathbf{y} := \mathbf{v}_1$ XOR $\mathbf{v}_3$), we have that $I(\mathbf{v}_1; \mathbf{y}) = I(\mathbf{v}_3; \mathbf{y}) = 0$. In this settings, $\mathbf{v}_1$ and $\mathbf{v}_2$ are mutually redundant for $\mathbf{y}$:

$$I(\mathbf{v}_1; \mathbf{y}|\mathbf{v}_2) = H(\mathbf{v}_1|\mathbf{v}_2) - H(\mathbf{v}_1|\mathbf{v}_2\mathbf{y}) = H(\mathbf{v}_1) - H(\mathbf{v}_1) = 0$$
$$I(\mathbf{v}_2; \mathbf{y}|\mathbf{v}_1) = H(\mathbf{v}_2|\mathbf{v}_1) - H(\mathbf{v}_2|\mathbf{v}_1\mathbf{y}) = H(\mathbf{v}_2) - H(\mathbf{v}_2) = 0$$

Analogously, $\mathbf{v}_2$ and $\mathbf{v}_3$ are also mutually redundant for $\mathbf{y}$ as the three random variables are not predictive for each other. Nevertheless, $\mathbf{v}_1$ and $\mathbf{v}_3$ are not mutually redundant for $\mathbf{y}$:

$$I(\mathbf{v}_1; \mathbf{y}|\mathbf{v}_3) = H(\mathbf{v}_1|\mathbf{v}_3) - \underbrace{H(\mathbf{v}_1|\mathbf{v}_3\mathbf{y})}_{0} = H(\mathbf{v}_1) = 1$$

$$I(\mathbf{v}_3; \mathbf{y}|\mathbf{v}_1) = H(\mathbf{v}_3|\mathbf{v}_1) - \underbrace{H(\mathbf{v}_3|\mathbf{v}_1\mathbf{y})}_{0} = H(\mathbf{v}_3) = 1$$

Where $H(\mathbf{v}_1|\mathbf{v}_3\mathbf{y}) = H(\mathbf{v}_3|\mathbf{v}_1\mathbf{y}) = 0$ follows from $\mathbf{v}_1 = \mathbf{v}_3$ XOR $\mathbf{y}$ and $\mathbf{v}_3 = \mathbf{v}_1$ XOR $\mathbf{y}$, while $H(\mathbf{v}_1) = H(\mathbf{v}_3) = 1$ holds by construction.

This counter-intuitive higher order interaction between multiple views makes our theory non-trivial to generalize to more than two views.

# E  EQUIVALENCES OF DIFFERENT OBJECTIVES

Different objectives in literature can be seen as a special case of the Multi-View Information Bottleneck principle. In this section we show that the supervised version of Information Bottleneck is equivalent to the corresponding Multi-View version whenever the two redundant views have only label information in common. A second subsection show equivalence between InfoMax and Multi-View Information Bottleneck whenever the two views are identical.

## E.1  MULTI-VIEW INFORMATION BOTTLENECK AND SUPERVISED INFORMATION BOTTLENECK

Whenever the two mutually redundant views $\mathbf{v}_1$ and $\mathbf{v}_2$ have only label information in common (or when one of the two views is the label itself) the Multi-View Information Bottleneck objective is equivalent to the respective supervised version. This can be shown by proving that $I(\mathbf{v}_1; \mathbf{z}_1|\mathbf{v}_2) = I(\mathbf{v}_1; \mathbf{z}_1|\mathbf{y})$, i.e. a representation $\mathbf{z}_1$ of $\mathbf{v}_1$ that is sufficient and minimal for $\mathbf{v}_2$ is also sufficient and minimal for $\mathbf{y}$.

**Proposition E.1.** *Let $\mathbf{v}_1$ and $\mathbf{v}_2$ be mutually redundant views for a label $\mathbf{y}$ that share only label information. Then a sufficient representation $\mathbf{z}_1$ of $\mathbf{v}_1$ for $\mathbf{v}_2$ that is minimal for $\mathbf{v}_2$ is also a minimal representation for $\mathbf{y}$.*

*Hypothesis:*

$(H_1)$ *$\mathbf{v}_1$ and $\mathbf{v}_2$ are mutually redundant for $\mathbf{y}$:* $I(\mathbf{v}_1; \mathbf{y}|\mathbf{v}_2) + I(\mathbf{v}_2; \mathbf{y}|\mathbf{v}_1) = 0$

$(H_2)$ *$\mathbf{v}_1$ and $\mathbf{v}_2$ share only label information:* $I(\mathbf{v}_1; \mathbf{v}_2) = I(\mathbf{v}_1; \mathbf{y})$

$(H_3)$ *$\mathbf{z}_1$ is sufficient for $\mathbf{v}_2$:* $I(\mathbf{v}_1; \mathbf{v}_2|\mathbf{z}_1) = 0$

*Thesis:*

$(T_1)$ $I(\mathbf{v}_1; \mathbf{z}_1|\mathbf{v}_2) = I(\mathbf{v}_1; \mathbf{z}_1|\mathbf{y})$

*Proof.*

1. Consider $I(\mathbf{v}_1; \mathbf{z})$:

$$
\begin{aligned}
I(\mathbf{v}_1; \mathbf{z}_1) &\overset{(P_3)}{=} I(\mathbf{v}_1; \mathbf{z}_1|\mathbf{v}_2) + I(\mathbf{v}_1; \mathbf{v}_2; \mathbf{z}_1) \\
&\overset{(P_3)}{=} I(\mathbf{v}_1; \mathbf{z}_1|\mathbf{v}_2) + I(\mathbf{v}_1; \mathbf{v}_2) - I(\mathbf{v}_1; \mathbf{v}_2|\mathbf{z}_1) \\
&\overset{(H_3)}{=} I(\mathbf{v}_1; \mathbf{z}_1|\mathbf{v}_2) + I(\mathbf{v}_1; \mathbf{v}_2) \\
&\overset{(H_1)}{=} I(\mathbf{v}_1; \mathbf{z}_1|\mathbf{v}_2) + I(\mathbf{v}_1; \mathbf{y})
\end{aligned}
$$

2. Using Corollary 1, from $(H_2)$ and $(H_3)$ follows $I(\mathbf{v}_1; \mathbf{y}|\mathbf{z}_1) = 0$

3. $I(\mathbf{v}_1; \mathbf{z})$ can be alternatively expressed as:

$$
\begin{aligned}
I(\mathbf{v}_1; \mathbf{z}_1) &\overset{(P_3)}{=} I(\mathbf{v}_1; \mathbf{z}_1|\mathbf{y}) + I(\mathbf{v}_1; \mathbf{y}; \mathbf{z}_1) \\
&\overset{(P_3)}{=} I(\mathbf{v}_1; \mathbf{z}_1|\mathbf{y}) + I(\mathbf{v}_1; \mathbf{y}) - I(\mathbf{v}_1; \mathbf{y}|\mathbf{z}_1) \\
&\overset{(Cor_1)}{=} I(\mathbf{v}_1; \mathbf{z}_1|\mathbf{y}) + I(\mathbf{v}_1; \mathbf{y})
\end{aligned}
$$

Equating 1 and 3, we conclude $I(\mathbf{v}_1; \mathbf{z}_1|\mathbf{v}_2) = I(\mathbf{v}_1; \mathbf{z}_1|\mathbf{y})$, therefore $\mathbf{z}_1$ which minimizes $I(\mathbf{v}_1; \mathbf{z}_1|\mathbf{v}_2)$ is also minimizing $I(\mathbf{v}_1; \mathbf{z}_1|\mathbf{y})$. When $I(\mathbf{v}_1; \mathbf{z}_1|\mathbf{y})$ is minimal, $I(\mathbf{y}; \mathbf{z}_1)$ is also minimal (see equation 2). $\qquad\square$

### E.2 MULTI-VIEW INFORMATION BOTTLENECK AND INFOMAX

Whenever $\mathbf{v}_1 = \mathbf{v}_2$, a representation $\mathbf{z}_1$ of $\mathbf{v}_1$ that is sufficient for $\mathbf{v}_2$ must contain all the original information regarding $\mathbf{v}_1$. Furthermore since $I(\mathbf{v}_1; \mathbf{z}_1|\mathbf{v}_2) = 0$ for every representation, no superfluous information can be identified and removed. As a consequence, a minimal sufficient representation $\mathbf{z}_1$ of $\mathbf{v}_1$ for $\mathbf{v}_2$ is any representation for which mutual information is maximal, hence InfoMax.

## F LOSS COMPUTATION

Starting from Equation 3, we consider the average of the losses $\mathcal{L}_1(\theta; \lambda_1)$ and $\mathcal{L}_2(\psi; \lambda_2)$ that aim to create the minimal sufficient representations $\mathbf{z}_1$ and $\mathbf{z}_2$ respectively:

$$
\mathcal{L}_{\frac{1+2}{2}}(\theta, \psi; \lambda_1, \lambda_2) = \frac{I_\theta(\mathbf{v}_1; \mathbf{z}_1|\mathbf{v}_2) + I_\psi(\mathbf{v}_2; \mathbf{z}_2|\mathbf{v}_1)}{2} + \frac{\lambda_1 I_\theta(\mathbf{v}_1; \mathbf{v}_2|\mathbf{z}_1) + \lambda_2 I_\psi(\mathbf{v}_1; \mathbf{v}_2|\mathbf{z}_1)}{2} \quad (6)
$$

Considering $\mathbf{z}_1$ and $\mathbf{z}_2$ on the same domain $\mathbb{Z}$, $I_\theta(\mathbf{v}_1; \mathbf{z}_1|\mathbf{v}_2)$ can be expressed as:

$$
\begin{aligned}
I_\theta(\mathbf{v}_1; \mathbf{z}_1|\mathbf{v}_2) &= \mathbb{E}_{\boldsymbol{v}_1, \boldsymbol{v}_2 \sim p(\mathbf{v}_1, \mathbf{v}_2)} \mathbb{E}_{\boldsymbol{z} \sim p_\theta(\mathbf{z}_1|\mathbf{v}_1)} \left[ \log \frac{p_\theta(\mathbf{z}_1 = \boldsymbol{z}|\mathbf{v}_1 = \boldsymbol{v}_1)}{p_\theta(\mathbf{z}_1 = \boldsymbol{z}|\mathbf{v}_2 = \boldsymbol{v}_2)} \right] \\
&= \mathbb{E}_{\boldsymbol{v}_1, \boldsymbol{v}_2 \sim p(\mathbf{v}_1, \mathbf{v}_2)} \mathbb{E}_{\boldsymbol{z} \sim p_\theta(\mathbf{z}_1|\mathbf{v}_1)} \left[ \log \frac{p_\theta(\mathbf{z}_1 = \boldsymbol{z}|\mathbf{v}_1 = \boldsymbol{v}_1)}{p_\psi(\mathbf{z}_2 = \boldsymbol{z}|\mathbf{v}_2 = \boldsymbol{v}_2)} \frac{p_\psi(\mathbf{z}_2 = \boldsymbol{z}|\mathbf{v}_2 = \boldsymbol{v}_2)}{p_\theta(\mathbf{z}_1 = \boldsymbol{z}|\mathbf{v}_2 = \boldsymbol{v}_2)} \right] \\
&= D_{\mathrm{KL}}(p_\theta(\mathbf{z}_1|\mathbf{v}_1)||p_\psi(\mathbf{z}_2|\mathbf{v}_2)) - D_{\mathrm{KL}}(p_\theta(\mathbf{z}_2|\mathbf{v}_1)||p_\psi(\mathbf{z}_2|\mathbf{v}_2)) \\
&\leq D_{\mathrm{KL}}(p_\theta(\mathbf{z}_1|\mathbf{v}_1)||p_\psi(\mathbf{z}_2|\mathbf{v}_2))
\end{aligned}
$$

Note that the bound is tight whenever $p_\psi(\mathbf{z}_2|\mathbf{v}_2)$ coincides with $p_\theta(\mathbf{z}_1|\mathbf{v}_2)$. This happens whenever $\mathbf{z}_1$ and $\mathbf{z}_2$ produce a consistent encoding. Analogously $I_\psi(\mathbf{v}_2; \mathbf{z}_2|\mathbf{v}_1)$ is upper bounded by $D_{\mathrm{KL}}(p_\psi(\mathbf{z}_2|\mathbf{v}_2)||p_\theta(\mathbf{z}_1|\mathbf{v}_1))$.

$I_\theta(\mathbf{v}_2; \mathbf{z}_1)$ can be rephrased as:

$$
\begin{aligned}
I_\theta(\mathbf{z}_1; \mathbf{v}_2) &\overset{(P_2)}{=} I_{\theta\psi}(\mathbf{z}_1; \mathbf{z}_2\mathbf{v}_2) - I_{\theta\psi}(\mathbf{z}_1; \mathbf{z}_2|\mathbf{v}_2) \\
&\overset{*}{=} I_{\theta\psi}(\mathbf{z}_1; \mathbf{z}_2\mathbf{v}_2) \\
&= I_{\theta\psi}(\mathbf{z}_1; \mathbf{z}_2) + I_{\theta\psi}(\mathbf{z}_1; \mathbf{v}_2|\mathbf{z}_2) \\
&\geq I_{\theta\psi}(\mathbf{z}_1; \mathbf{z}_2)
\end{aligned}
$$

Where $^*$ follows from $\mathbf{z}_2$ representation of $\mathbf{v}_2$. The bound reported in this equation is tight whenever $\mathbf{z}_2$ is sufficient for $\mathbf{z}_1$ ($I_{\theta\psi}(\mathbf{z}_1; \mathbf{v}_2|\mathbf{z}_2) = 0$). This happens whenever $\mathbf{z}_2$ contains all the information regarding $\mathbf{z}_1$ (and therefore $\mathbf{v}_1$). Once again, the same bound can symmetrically be used to show $I_\theta(\mathbf{z}_2; \mathbf{v}_1) \geq I_{\theta\psi}(\mathbf{z}_1; \mathbf{z}_2)$. Therefore, the loss function in Equation 6 can be upper-bounded with:

$$\mathcal{L}_{\frac{1+2}{2}}(\theta, \psi; \lambda_1, \lambda_2) \leq D_{SKL}(p_\theta(\mathbf{z}_1|\mathbf{v}_1)||p_\psi(\mathbf{z}_2|\mathbf{v}_2)) - \frac{\lambda_1 + \lambda_2}{2} I_{\theta\psi}(\mathbf{z}_1; \mathbf{z}_2) \tag{7}$$

Where:

$$D_{SKL}(p_\theta(\mathbf{z}_1|\mathbf{v}_1)||p_\psi(\mathbf{z}_2|\mathbf{v}_2)) := \frac{1}{2} D_{\mathrm{KL}}(p_\theta(\mathbf{z}_1|\mathbf{v}_1)||p_\psi(\mathbf{z}_2|\mathbf{v}_2)) + \frac{1}{2} D_{\mathrm{KL}}(p_\psi(\mathbf{z}_2|\mathbf{v}_2)||p_\theta(\mathbf{z}_1|\mathbf{v}_1))$$

Lastly, multiplying both terms with $\beta := \frac{2}{\lambda_1 + \lambda_2}$ and re-parametrizing the objective, we obtain:

$$\mathcal{L}_{MIB}(\theta, \psi; \beta) = -I_{\theta\psi}(\mathbf{z}_1; \mathbf{z}_2) + \beta\, D_{SKL}(p_\theta(\mathbf{z}_1|\mathbf{v}_1)||p_\psi(\mathbf{z}_2|\mathbf{v}_2)) \tag{8}$$

# G  EXPERIMENTAL PROCEDURE AND DETAILS

## G.1  MODELING

The two stochastic encoders $p_\theta(\mathbf{z}_1|\mathbf{v}_1)$ and $p_\psi(\mathbf{z}_2|\mathbf{v}_2)$ are modeled by Normal distributions parametrized with neural networks $(\boldsymbol{\mu}_\theta, \boldsymbol{\sigma}_\theta^2)$ and $(\boldsymbol{\mu}_\psi, \boldsymbol{\sigma}_\psi^2)$ respectively:

$$p_\theta(\mathbf{z}_1|\mathbf{v}_1) := \mathcal{N}\left(\mathbf{z}_1|\boldsymbol{\mu}_\theta(\mathbf{v}_1), \boldsymbol{\sigma}_\theta^2(\mathbf{v}_1)\right)$$
$$p_\psi(\mathbf{z}_2|\mathbf{v}_2) := \mathcal{N}\left(\mathbf{z}_2|\boldsymbol{\mu}_\psi(\mathbf{v}_2), \boldsymbol{\sigma}_\psi^2(\mathbf{v}_2)\right)$$

Since the density of the two encoders can be evaluated, the symmetrized KL-divergence in equation 4 can be directly computed. On the other hand, $I_{\theta\psi}(\mathbf{z}_1; \mathbf{z}_2)$ requires the use of a mutual information estimator.

To facilitate the optimization, the hyper-parameter $\beta$ is slowly increased during training, starting from a small value $\approx 10^{-4}$ to its final value with an exponential schedule. This is because the mutual information estimator is trained together with the other architectures and, since it starts from a random initialization, it requires an initial warm-up. Starting with bigger $\beta$ results in the encoder collapsing into a fixed representation. The update policy for the hyper-parameter during training has not shown strong influence on the representation, as long as the mutual information estimator network has reached full capacity.

All the experiments have been performed using the Adam optimizer with a learning rate of $10^{-4}$ for both encoders and the estimation network. Higher learning rate can result in instabilities in the training procedure. The results reported in the main text relied on the Jensen-Shannon mutual information estimator (Devon Hjelm et al., 2019) since the InfoNCE counterpart (van den Oord et al., 2018) generally resulted in worse performance that could be explained by the effect of the factorization of the critic network (Poole et al., 2019).

## G.2  SKETCHY EXPERIMENTS

- **Input:** The two views for the sketch-based classification task consist of 4096 dimensional sketch and image features extracted from two distinct VGG-16 network models which were pre-trained on images and sketches from the TU-Berlin dataset Eitz et al. (2012) for end-to-end classification. The feature extractors are frozen during the training procedure of for the two representations. Each training iteration used batches of size $B = 128$.

- **Encoder and Critic architectures:** Both sketch and image encoders consist of multi-layer perceptrons of 2 hidden ReLU units of size 2,048 and 1,024 respectively with an output of size 2x64 that parametrizes mean and variance for the two Gaussian posteriors. The critic architecture also consists of a multi layer perceptron of 2 hidden ReLU units of size 512.

- **$\beta$ update policy:** The initial value of $\beta$ is set to $10^{-4}$. Starting from the 10,000[th] training iteration, the value of $\beta$ is exponentially increased up to 1.0 during the following 250,000 training iterations. The value of $\beta$ is then kept fixed to one until the end of the training procedure (500,000 iterations).

- **Evaluation:** All natural images are used as both training sets and retrieval galleries. The 64 dimensional real outputs of sketch and image representation are compared using Euclidean distance. For having a fair comparison other methods that rely on binary hashing (Liu et al., 2017; Zhang et al., 2018), we used Hamming distance on a binarized representation (obtained by applying iterative quantization Gong et al. (2013) on our real valued representation). We report the mean average precision (mAP@all) and precision at top-rank 200 (Prec@200) Su et al. (2015) on both the real and binary representation to evaluate our method and compare it with prior works.

## G.3 MIR-Flickr Experiments

Figure 6: Examples of pictures $\mathbf{v}_1$, tags $\mathbf{v}_2$ and category labels $\mathbf{y}$ for the MIR-Flickr dataset (Srivastava & Salakhutdinov, 2014). As visualized is the second row, the tags are not always predictive of the label. For this reason, the mutual redundancy assumption holds only approximately.

| $\mathbf{v}_1 \in \mathbb{R}^{3857}$ | $\mathbf{v}_2 \in \{0,1\}^{2000}$ | $\mathbf{y} \in \{0,1\}^{38}$ |
| --- | --- | --- |
| | "watermelon", "hilarious", "chihuahua", "dog" | "animals", "dog", "food" |
| | "colors", "cores", "centro", "comercial", "building" | "clouds", "sky", "structures" |

- **Input:** Whitening is applied to the handcrafted image features. Batches of size $B = 128$ are used for each update step.

- **Encoders and Critic architectures:** The two encoders consists of a multi layer perceptron of 4 hidden ReLU units of size 1,024, which exactly resemble the architecture used in Wang et al. (2016). Both representations $\mathbf{z}_1$ and $\mathbf{z}_2$ have a size of 1,024, therefore the two architecture output a total of 2x1,024 parameters that define mean and variance of the respective factorized Gaussian posterior. Similarly to the Sketchy experiments, the critic is consists of a multi-layer perceptron of 2 hidden ReLU units of size 512.

- $\beta$ **update policy:** The initial value of $\beta$ is set to $10^{-8}$. Starting from $150000^{\text{th}}$ iteration, $\beta$ is set to exponentially increase up to 1.0 (and $10^{-3}$) during the following 150,000 iterations.

- **Evaluation:** Once the models are trained on the *unlabeled* set, the representation of the 25,000 *labeled* images is computed. The resulting vectors are used for training and evaluating a multi-label logistic regression classifier on the respective splits. The optimal parameters (such as $\beta$) for our model are chosen based on the performance on the validation set. In Table 3, we report the aggregated mean of the 5 test splits as the final value mean average precision value.

## G.4 MNIST Experiments

- **Input:** The two views $\mathbf{v}_1$ and $\mathbf{v}_2$ for the MNIST dataset are generated by applying small translation ([0-10]%), rotation ([-15,15] degrees), scale ([90,110]%), shear ([-15,15] degrees) and pixel corruption (20%). Batches of size $B = 64$ samples are used during training.

- **Encoders, Decoders and Critic architectures:** All the encoders used for the MNIST experiments consist of neural networks with two hidden layers of 1,024 units and ReLU activations, producing a 2x64-dimensional parameter vector that is used to parameterize mean and variance for the Gaussian posteriors. The decoders used for the VAE experiments also consist of the networks of the same size. Similarly, the critic architecture used for mutual information estimation consists of two hidden layers of 1,204 units each and ReLU activations.

- $\beta$ **update policy:** The initial value of $\beta$ is set to $10^{-3}$, which is increased with an exponential schedule starting from the 50,000[th] until 1the 50,000[th] iteration. The value of $\beta$ is then kept constant until the 1,000,000[th] iteration. The same annealing policy is used to trained the different $\beta$-VAEs reported in this work.

- **Evaluation:** The trained representation are evaluated following the well-known protocol described in Tschannen et al. (2019); Tian et al. (2019); Bachman et al. (2019); van den Oord et al. (2018). Each logistic regression is trained 5 different balanced splits of the training set for different percentages of training examples, ranging from 1 example per label to the whole training set. The accuracy reported in this work has been computed on the disjoint test set. Mean and standard deviation are computed according to the 5 different subsets used for training the logistic regression. Mean and variance for the mutual information estimation reported on the Information Plane (Figure 4) are computed by training two estimation networks from scratch on the final representation of the non-augmented train set. The two estimation architectures consist of 2 hidden layers of 2048 and 1024 units each, and have been trained with batches of size $B = 256$ for a total of approximately 25,000 iterations. The Jensen-Shannon mutual information lower bound is maximized during training, while the numerical estimation are computed using an energy-based bound (Poole et al., 2019; Devon Hjelm et al., 2019). The final values for $I(\mathbf{x}; \mathbf{z})$ and $I(\mathbf{y}; \mathbf{z})$ are computed by averaging the mutual information estimation on the whole dataset. In order to reduce the variance of the estimator, the lowest and highest 5% are removed before averaging. This practical detail makes the estimation more consistent and less susceptible to numerical instabilities.

### G.4.1 RESULTS AND VISUALIZATION

In this section we include additional quantitative results and visualizations which refer to the single-view MNIST experiments reported in section 5.2.

Table 2 reports the quantitative results used for to produce the visualizations reported in Figure 4, including the comparison between the performance resulting from different mutual information estimators. As the Jensen-Shannon estimator generally resulted in better performance for the InfoMax, MV-InfoMax and MIB models, all the experiments reported on the main text make use of this estimator. Note that the InfoMax model with the $I_{\text{JS}}$ estimator is equivalent to the global model reported in Devon Hjelm et al. (2019), while MV-InfoMax with the $I_{\text{NCE}}$ estimator results in a similar architecture to the one introduced in Tian et al. (2019).

| Model | $I(\mathbf{x}; \mathbf{z})$ [nats] | $I(\mathbf{z}; \mathbf{y})$ [nats] | Test Accuracy [%] | | | |
|---|---|---|---|---|---|---|
| | | | 10 Ex | 50 Ex | 3750 Ex | 60000 Ex |
| VAE (beta=0) | $12.5 \pm 0.7$ | $2.3 \pm 0.2$ | $43.8 \pm 1.6$ | $65.6 \pm 3.3$ | $89.0 \pm 0.4$ | $91.3 \pm 0.1$ |
| VAE (beta=4) | $7.5 \pm 1.0$ | $2.0 \pm 0.2$ | $55.9 \pm 2.6$ | $81.4 \pm 4.0$ | $94.2 \pm 0.3$ | $96.0 \pm 0.2$ |
| VAE (beta=8) | $3.0 \pm 0.5$ | $1.0 \pm 0.1$ | $43.8 \pm 2.8$ | $61.1 \pm 4.8$ | $81.9 \pm 1.1$ | $87.2 \pm 0.6$ |
| InfoMax ($I_{\text{NCE}}$) | $12.8 \pm 0.5$ | $2.3 \pm 0.2$ | $25.4 \pm 1.9$ | $39.6 \pm 3.3$ | $69.2 \pm 0.7$ | $74.6 \pm 0.6$ |
| InfoMax ($I_{\text{JS}}$) | $13.7 \pm 0.7$ | $2.2 \pm 0.2$ | $35.0 \pm 2.8$ | $52.5 \pm 2.8$ | $74.4 \pm 1.1$ | $78.2 \pm 1.2$ |
| MV-InfoMax ($I_{\text{NCE}}$) | $12.2 \pm 0.7$ | $2.3 \pm 0.2$ | $50.2 \pm 3.6$ | $75.8 \pm 3.8$ | $94.6 \pm 0.4$ | $96.5 \pm 0.1$ |
| MV-InfoMax ($I_{\text{JS}}$) | $11.1 \pm 1.0$ | $2.3 \pm 0.2$ | $54.0 \pm 6.1$ | $78.3 \pm 4.4$ | $94.1 \pm 0.3$ | $95.90 \pm 0.08$ |
| MIB ($\beta = 1$, $I_{\text{NCE}}$) | $4.6 \pm 0.7$ | $2.1 \pm 0.2$ | $81.8 \pm 5.0$ | $92.7 \pm 0.9$ | $97.19 \pm 0.08$ | $97.75 \pm 0.05$ |
| MIB ($\beta = 1$, $I_{\text{JS}}$) | $2.4 \pm 0.2$ | $2.1 \pm 0.2$ | $\mathbf{97.1 \pm 0.2}$ | $\mathbf{97.2 \pm 0.2}$ | $\mathbf{97.70 \pm 0.06}$ | $\mathbf{97.82 \pm 0.01}$ |

Table 2: Comparison of the amount of input information $I(\mathbf{x}; \mathbf{z})$, label information $I(\mathbf{z}; \mathbf{y})$, and accuracy of a linear classifier trained with different amount of labeled Examples (Ex) for the models reported in Figure 4. Both the results obtained using the Jensen-Shannon $I_{\text{JSD}}$ (Devon Hjelm et al., 2019; Poole et al., 2019) and the InfoNCE $I_{\text{NCE}}$ (van den Oord et al., 2018) estimators are reported.

Figure 7 reports the linear projection of the embedding obtained using the MIB model. The latent space appears to roughly consists of ten clusters which corresponds to the different digits. This observation is consistent with the empirical measurement of input and label information $I(\mathbf{x}; \mathbf{z}) \approx I(\mathbf{z}; \mathbf{y}) \approx \log 10$, and the performance of the linear classifier in scarce label regimes. As the cluster are distinct and concentrated around the respective centroids, 10 labeled examples are sufficient to align the centroid coordinates with the digit labels.

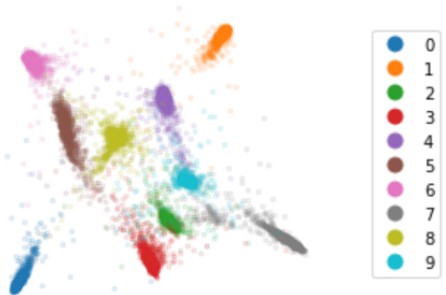

Figure 7: Linear projection of the embedding obtained by applying the MIB encoder to the MNIST test set. The 64 dimensional representation is projected onto the two principal components. Different colors are used to represent the 10 digit classes.

# H  ABLATION STUDIES

## H.1  DIFFERENT RANGES OF DATA AUGMENTATION

Figure 8 visualizes the effect of different ranges of corruption probabily as data augmentation strategy to produce the two views $\mathbf{v}_1$ and $\mathbf{v}_2$. The MV-InfoMax Model does not seem to get any advantage from the use increasing amount of corruption, and it representation remains approximately in the same region of the information plane. On the other hand, the models trained with the MIB objective are able to take advantage of the augmentation to remove irrelevant data information and the representation transitions from the top right corner of the Information Plane (no-augmentation) to the top-left. When the amount of corruption approaches 100%, the mutual redundancy assumption is clearly violated, and the performances of MIB deteriorate. In the initial part of the transitions between the two regimes (which corresponds to extremely low probability of corruption) the MIB models drops some label information that is quickly re-gained when pixel corruption becomes more frequent. We hypothesize that this behavior is due to a problem with the optimization procedure, since the corruption are extremely unlikely, the Monte-Carlo estimation for the symmetrized Kullback-Leibler divergence is more biased. Using more examples of views produced from the same data-point within the same batch could mitigate this issue.

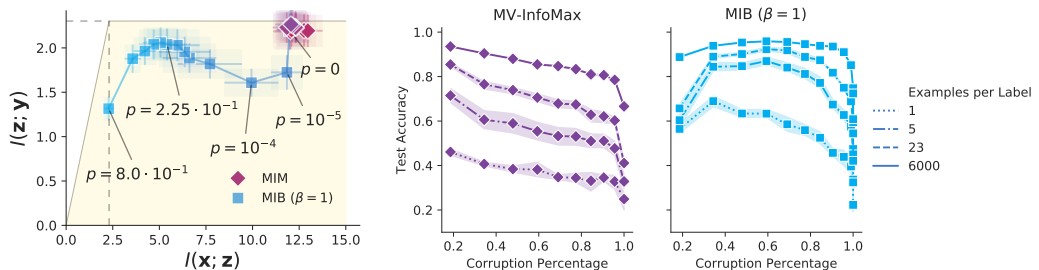

Figure 8: Visualization of the coordinates on the Information Plane (plot on the left) and prediction accuracy (center and right) for the MV-InfoMax and MIB objectives with different amount of training labels and corruption percentage used for data-augmentation.

## H.2  EFFECT OF $\beta$

The hyper-parameter $\beta$ (Equation 5) determines the trade-off between sufficiency and minimality of the representation for the second data view. When $\beta$ is zero, the training objective of MIB is equiv-

alent to the Multi-View InfoMax target, since the representation has no incentive to discard any information. When $0 < \beta \leq 1$ the sufficiency constrain is enforced, while the superfluous information is gradually removed from the representation. Values of $\beta > 1$ can result in representations that violate the sufficiency constraint, since the minimization of $I(\mathbf{x}; \mathbf{z}|\mathbf{v}_2)$ is prioritized. The trade-off resulting from the choice of different $\beta$ is visualized in Figure 9 and compared against $\beta$-VAE. Note that in each point of the pareto-front the MIB model results in a better trade-off between $I(\mathbf{x}; \mathbf{z})$ and $I(\mathbf{y}; \mathbf{z})$ when compared to $\beta$-VAE. The effectiveness of the Multi-View Information Bottleneck model is also justified by the corresponding values of predictive accuracy.

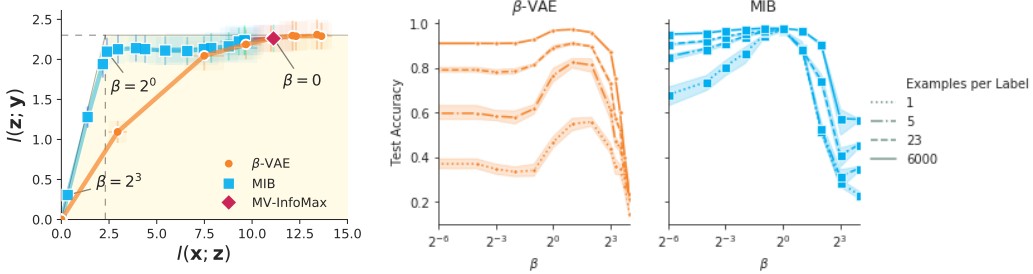

Figure 9: Visualization of the coordinates on the Information Plane (plot on the left) and prediction accuracy (center and right) for the $\beta$-VAE, Multi-View InfoMax and Multi-View Information Bottleneck objectives with different amount of training labels and different values of the respective hyperparameter $\beta$.

