# OpenReview forum: "Learning Robust Representations via Multi-View Information Bottleneck"
_ICLR.cc/2020/Conference — Accept (Poster)_

### Official Review · AnonReviewer2 · 2019-10-23
**Official Blind Review #2**

**Rating:** 8

**Review:**

This is a good multiview representation learning paper with new insights. The authors propose to learn variables z_1 and z_2, which are consistent, contain view-invariant information but discard as much view-specific information as possible.
The paper relies on mutual information estimation and is reconstruction-free. It is mentioned in some previous works (e.g. Aaron van den Oord et al. 2018), that reconstruction loss can introduce bias that has a negative effect on the learned representation.
Comparing to existing multiview representation learning approaches that try to maximize the mutual information between learned representation and the view(s), this paper clearly defines superfluous information that we should try to throw away and figure out how to obtain sufficiency learned representation for output. The authors also draw clear connections between a few existing (multiview) representation learning methods to their proposed approaches.
The experimental results on the right side of Figure 3, deliver a very interesting conclusion. In low-resource case, robust feature (obtained by using the larger beta, discarding more superfluous information) is crucial for achieving good performance. While when the amount of labeled data samples is enough, vice-versa.

Here are my major concerns:
1.	In the paper, the authors said the original formulation of IB is only applicable to supervised learning. That is true, but the variational information bottleneck paper [Alexander A. Alem et al. 2017] already showed the connection of unsupervised VIB to VAE in the appendix.
2.	I would not consider the data augmentation used to extend single-view data to “pseudo-multiview” as a contribution. This has been done before (e.g. in the multiview MNIST experiment part of the paper "On Deep Multi-View Representation Learning").
3.	Which MV-InfoMax do you really compare to? You listed a few of them: (Ji et al., 2019; Henaff et al., ´ 2019; Tian et al., 2019; Bachman et al., 2019) in the related work section.
4.	I think the authors should also make a more careful claim on their results in MIR-Flickr.
I’d rather not saying MIB generally outperforms MV-InfoMax on MIR-Flickr, as MIB does not (clearly) outperform MV-InfoMax when enough labeled data is available for training downstream recognizers. But MIB does clearly outperform MV-InfoMax when scaling down the percentage of labeled samples used.
5.	Regarding baselines/experiments
a.	In Figure 4, it seems that VAE (with beta=4) outperforms MV-InfoMax. Why the ``"pseudo-second view" does not help Mv-Infomax in this scenario? Why VAE is clearly better than Infomax?
b.	In Figure 3, you might also tune beta for VCCA and its variants, like what you did for VAE/VIB in a single view.
6.	Do you think your approach can be extended to more than two views easily?
For me, it seems the extension is not trivial, as it requires o(n^2) terms in your loss for n views.
But this is minor.



**Experience Assessment:**

I have published one or two papers in this area.

**Review Assessment: Checking Correctness Of Derivations And Theory:**

I carefully checked the derivations and theory.

**Review Assessment: Checking Correctness Of Experiments:**

I carefully checked the experiments.

**Review Assessment: Thoroughness In Paper Reading:**

I read the paper at least twice and used my best judgement in assessing the paper.

---

> ### Author Response · Authors · 2019-11-15
> **Response to Reviewer 2 (Part 1)**
>
> 1) In the paper, the authors said the original formulation of IB is only applicable to supervised learning. That is true, but the variational information bottleneck paper [Alexander A. Alem et al. 2017] already showed the connection of unsupervised VIB to VAE in the appendix.
>
> In supervised settings, VIB allows one to create a representation that discards irrelevant input information. The unsupervised extension of VIB mentioned in [Alemi et al. (2017)] is equivalent to the beta-VAE model, in which the beta hyper-parameter regulates the trade-off between distortion and rate [Alemi et al. (2018)]. In this setting, however, we have no guarantees that the information discarded by the model is irrelevant for the task. Our model, on the other hand, makes use of a source of redundant information (unsupervised multi-view setting) to create a representation that discards only irrelevant information.
> We updated our claims in the introduction to clarify the distinction between the three models.
>
> 2) I would not consider the data augmentation used to extend single-view data to “pseudo-multiview” as a contribution. This has been done before (e.g. in the multiview MNIST experiment part of the paper "On Deep Multi-View Representation Learning").
>
> We updated claim (3) in the introduction to clarify that our contribution does not consist in the definition of “pseudo-multiview” but rather in connecting the well-known data augmentation procedure to the mutual redundancy condition introduced in this work.
> We believe that this connection could be useful as it defines some constraints on the function class used for augmentation, which can be expressed and described using information-theoretic quantities.
>
>  3) Which MV-InfoMax do you really compare to? You listed a few of them: (Ji et al., 2019; Henaff et al., ´ 2019; Tian et al., 2019; Bachman et al., 2019) in the related work section.
>
> The version of MV-InfoMax reported in the experiments is based on Tian et al (2019), with the only difference that we used an alternative mutual information estimator. The InfoNCE estimator used in [Tian et al (2019)] allows for faster computation but usually results in slightly worse estimations than the Jensen-Shannon estimator [Poole 2018]. For this reason, we decided to consistently use the Jensen-Shannon estimator for the rest of the experiments as it results in better performance for all three models.
>
> We added appendix G.4.1 to report the performance and mutual information estimation obtained by using both InfoNCE and Jensen-Shannon estimators for InfoMax, MV-InfoMax, and MIB. We also added a few sentences in the experimental section to clarify our mutual information estimation choice.
>
> 4) I think the authors should also make a more careful claim on their results in MIR-Flickr. I’d rather not saying MIB generally outperforms MV-InfoMax on MIR-Flickr, as MIB does not (clearly) outperform MV-InfoMax when enough labeled data is available for training downstream recognizers. But MIB does clearly outperform MV-InfoMax when scaling down the percentage of labeled samples used.
>
> We updated our claim in the abstract of the paper to clarify this point by specifically noting that we provide state-of-the-art results only in the label-limited regime.

---

> ### Author Response · Authors · 2019-11-15
> **Response to Reviewer 2 (Part 2)**
>
> 5.a) In Figure 4, it seems that VAE (with beta=4) outperforms MV-InfoMax.
> i) Why the "pseudo-second view" does not help Mv-Infomax in this scenario?
>
> At inference time both the VAE and MV-InfoMax models have access to only one view.
> For this reason, MV-InfoMax can exploit the second view only to identify which features of the input are kept and which ones are discarded at training time. As mentioned in Section 4, MV-InfoMax has no incentive to discard any information since any representation that contains at least the information that the two views have in common is equally optimal according to its training objective. Empirically, the representation obtained with MV-InfoMax contains a lot of superfluous information (~11 nats for MV-InfoMax and ~14 nats for InfoMax ), which can justify their reduced performance.
>
> The Variational Autoencoder model, on the other hand, makes use of a compression term that is regulated by the hyper-parameter beta. The compression is completely agnostic of the label and we have no guarantees that the label information is retained, but, in practice, for any beta<=4 most of the label information is kept (~2 nats). A VAE trained with beta=4 produces a representation which contains most of the label information but is less influenced by other variations in the input (~7 nats) when compared to InfoMax and MV-InfoMax. We hypothesize that the slightly better performance of the VAE model can be explained by the interplay between the amount of label and superfluous information in the representations.
> Numerical measurements of accuracy and mutual information have been added to Table 2 in the appendix to facilitate the comparison between the different models.
>
> ii) Why VAE is clearly better than Infomax?
>
> The training objective of VAE (beta=0) and InfoMax are equivalent in their goal of maximizing the amount of information included in the representation. We believe that the difference between their performance is due to the use of different strategies to maximize the same quantity. As reported in [McAllester & Stratos (2018)] estimators based on lower-bounds of a Kullback-Leibler divergence (as the Jensen-Shannon or InfoNCE estimator used for the InfoMax models) are generally worse than the ones based on difference of entropies (as VAE with beta=0 ) when estimating high values of Mutual Information (i.e. the mutual information between the input and the representation). The same estimator works better for the MIB model since the amount of information to estimate and maximize is lower (i.e. the shared information across the two views).
>
> 5.b) In Figure 3, you might also tune beta for VCCA and its variants, like what you did for VAE/VIB in a single view.
>
> The models based on VCCA have multiple parameters which play the same role as beta (weights for each of the each of the two cross-modal and uni-modal reconstruction terms).  The interplay between these parameters is complicated and so we chose not to try to explore this space since we weren't sure what insight it would provide, and we didn't expect competitive results given that the beta=1 version of our model when trained with only 2% of the labels was already outperforming the best version of VCCA when it was trained with all of the data.
>
> 6) Do you think your approach can be extended to more than two views easily? For me, it seems the extension is not trivial, as it requires o(n^2) terms in your loss for n views.
>
> We addressed this above as part of shared question (2)

---

### Official Review · AnonReviewer1 · 2019-10-24
**Official Blind Review #1**

**Rating:** 8

**Review:**

In this paper, the authors extend the Information Bottleneck method (to build robust representations by removing information unrelated to the target labels) to the unsupervised setting. Since label information is not available in this setting, the authors leverage multi-view information (e.g., using two images of the same object) , which requires assuming that both views contain all necessary information for the subsequent label prediction task. The representation should then focus on capturing the information shared by both views and discarding the rest. A loss function for learning such representations is proposed. The effectiveness of the proposed technique is confirmed on two datasets. It is also shown to work when doing data augmentation with a single view.

Overall the paper is well motivated, well placed in the literature and well written. Mathematical derivations are provided. Experimental methodology follows the existing literature, seem reasonable and results are convincing. I do not have major negative comments for the authors. This is however not my research area and have only a limited knowledge of the existing body of work.

Comments/Questions:
- How limiting is the multi-view assumption? Are there well-known cases where it doesn't hold? I feel it would be hard to use, say, with text. Has this been discussed in the literature? Some pointers or discussion would be interesting.
- Sketchy dataset: Could the DSH algorithm (one of the best prior results) be penalized by not using the same feature extractor you used?
- Sketchy dataset: Can a reference for the {Siamese,Triplet}-AlexNet results be provided?
- Sketchy dataset: for reproducibility, what is the selected \beta?
- I find it very hard to believe that the accuracy stays constant no matter the number of examples per label used. How can an encoder be trained on 10 images? Did I misunderstand the meaning of this number? Can this be clarified?
- Again for reproducibility, listing the raw numbers for the MNIST experiments would be nice.
- If I understood the experiments correctly, "scarce label regime" is used for both the MIR-Flickr and MNIST datasets, meaning two different things (number of labels per example vs number of examples per label), which is slightly confusing.

Typos:
Page 1: it's -> its
Page 6: the the -> the
Page 7: classifer -> classifier
Page 8: independently -> independent


**Experience Assessment:**

I do not know much about this area.

**Review Assessment: Checking Correctness Of Derivations And Theory:**

I assessed the sensibility of the derivations and theory.

**Review Assessment: Checking Correctness Of Experiments:**

I assessed the sensibility of the experiments.

**Review Assessment: Thoroughness In Paper Reading:**

I read the paper at least twice and used my best judgement in assessing the paper.

---

> ### Author Response · Authors · 2019-11-15
> **Response to Reviewer 1**
>
> 1) How limiting is the multi-view assumption? Are there well-known cases where it doesn't hold? I feel it would be hard to use, say, with text. Has this been discussed in the literature? Some pointers or discussion would be interesting.
>
> We answered this above as part of shared question (1)
>
> 2) Sketchy dataset: Could the DSH algorithm (one of the best prior results) be penalized by not using the same feature extractor you used?
>
> The DSH algorithm may be limited by their use of an AlexNet instead of a VGG network, but they also fine-tune the AlexNet as part of their model, while our model directly uses the features unmodified.  So it's unclear how these two changes affect the performance on the balance.  We were unable to produce an updated version of their results with a VGG network because they do not provide code or hyper-parameter settings for training their models, or details on how they did the pretraining.  In order to facilitate future comparison, we will release the preprocessed dataset with the 5 splits used for our experiments upon paper acceptance.
>
>
>
> 3) Sketchy dataset: Can a reference for the {Siamese,Triplet}-AlexNet results be provided? For reproducibility, what is the selected \beta?
>
> A reference for {Siamese,Triplet}-AlexNet results has been added to Table 1 together with the selected value of beta for MIB.
>
> 4) I find it very hard to believe that the accuracy stays constant no matter the number of examples per label used. How can an encoder be trained on 10 images? Did I misunderstand the meaning of this number? Can this be clarified?
>
> The MNIST experiments are performed with the following procedure:
>
> i) Using the whole training set without any labels (i.e. unsupervised) we train the encoder network which generates a representation for each input picture.  We then freeze the weight of the encoder network.
>
> ii) Then the randomly chosen set of training labels are used to train a linear classifier to map from encoded samples to the labels.  Note here that in our experiments, the chosen amount of labels does indeed range from one example per label, up to and including all labels.
>
> iii) Each classifier is evaluated on a disjoint test set. This is done by first encoding the unseen test set using the encoder trained in i) and plugging the representations into the linear classifier trained in ii)
> In this setting, the encoder has access to the whole (unlabeled) training set, but the classifier is trained using different amounts of examples.
> The MIB model produces a representation that contains approximately 2.3 nats (Figure 4) which corresponds roughly to the amount of information associated with a categorical distribution on 10 classes with uniform probability (ln10 ~2.3 nats). Further investigating this interesting property, we visualized the representation produced by our model by projecting the representation of the test set on the 2 principal components (Figure 7 added in Appendix G.4.1). The representation for the test digits roughly consists of 10 linearly separable clusters, which explains why 10 examples are sufficient to align cluster centroids and labels.
>
> 5) Again for reproducibility, listing the raw numbers for the MNIST experiments would be nice.
>
> We added appendix G.4.1 in which we report accuracy for different numbers of examples and mutual information estimation corresponding to the comparison reported in Figure 4.
>
> 6) If I understood the experiments correctly, "scarce label regime" is used for both the MIR-Flickr and MNIST datasets, meaning two different things (number of labels per example vs number of examples per label), which is slightly confusing.
>
> By “scarce label regime”, we mean a reduced number of labeled examples.
> This translates into slightly different settings for single-label (MNIST) and multi-label (Flickr) classification problems.
> In both cases, we simulate the lack of labels by picking a subset of labeled examples with the same distribution p(x,y) as the original training set.
> Since the label distribution on MNIST is uniform, we generate training subsets by picking the same number of examples for each class.
> The same procedure is not possible for the Flickr experiments as the label distribution is uneven and each example has multiple labels. For this reason, we uniformly subsample the original training set without replacement.
> The x-axis in Figure 3 and 4 have been updated to consistently report the number of labeled examples used for training a classifier on top of the specified representation.
>
> 7) Typos
>
> We thank the reviewer for identifying the mistakes, which have been fixed in the current version of the paper.

---

### Official Review · AnonReviewer3 · 2019-10-29
**Official Blind Review #3**

**Rating:** 6

**Review:**

This paper extends the information bottleneck method of Tishby et al. (2000) to the unsupervised setting. By taking advantage of multi-view data, they provide two views of the same underlying entity.  Experimetal results on two standard multi-view datasets validate the efficacy of the proposed method.
I have three questions about this work.
1. The proposed method only provides two views of the same underlying entity, what about 3 or more views?
2. Can this method be used for multi-modality case?
3. What about the time efficiency of the proposed method?

**Experience Assessment:**

I have published in this field for several years.

**Review Assessment: Checking Correctness Of Derivations And Theory:**

I carefully checked the derivations and theory.

**Review Assessment: Checking Correctness Of Experiments:**

I carefully checked the experiments.

**Review Assessment: Thoroughness In Paper Reading:**

I read the paper thoroughly.

---

> ### Author Response · Authors · 2019-11-15
> **Response to Reviewer 3**
>
> 1) The proposed method only provides two views of the same underlying entity, what about 3 or more views?
>
> We addressed this above as part of shared question (2)
>
> 2) Can this method be used for multi-modality case?
>
> We addressed this above as part of shared question (1)
>
> 3) What about the time efficiency of the proposed method?
>
> The proposed MIB architecture involves training 2 encoders and one auxiliary architecture as in [Tian et al. (2019), Bachman et al. (2019)].  Such models typically train faster than the ones that involve the use of decoders (e.g. VAE, MVAE, VCCA), or involve more complicated architectures consisting of multiple modules (e.g. GDH, DSH). On the other hand, since the beta hyper-parameter has to be slowly increased during training to ensure stability, the total number of training steps required for convergence is slightly bigger than InfoMax and MV-InfoMax. Both training time per epoch and the total number of training steps for convergence mostly depend on the specific choice of mutual information estimation and corresponding auxiliary neural network architecture and is a current subject of exploration in recent work [Poole et al. (2019), Belghazi et al. (2018)].

---

### Author Response · Authors · 2019-11-15
**Shared Response to Reviewers**

We thank the reviewers for their useful feedback and comments.  The following addresses questions asked by multiple reviewers:

Shared Question 1: Applicability of Multi-View Information Bottleneck and mutual redundancy assumption
(Addresses Reviewer 1's first question and Reviewer 3's second question)

The multi-view literature has explored and discussed the applicability of the assumption that each view is sufficient for correct classification [Zhao (2017)]. They find that this assumption holds in a wide variety of circumstances resulting in a large community exploring this space.  Our mutual redundancy assumption is weaker than the standard multi-view assumption, requiring only that the two views are "equally certain" about the label allowing it to be applied in an even wider set of circumstances. Specifically, it can be applied in multi-modal settings, such as the MIR-Flickr and SBIR (Sketchy dataset) settings shown in our experiments, as well as text-based tasks such as translation and paraphrasing where both text views contain the same relevant (semantic) information. Two additional points are also worth noting:

1) Our method can be applied even if the mutual redundancy assumption holds only approximately (I(v_1;y|v_2)+I(v_2;y|v_1)<epsilon), by using a lower value of beta, to reduce the pressure to remove all information which is not shared. We can see this empirically with the Flickr dataset where the mutual redundancy constraint is clearly violated since some of the tags are not as predictive of the labels as the full image (see Figure 6 in appendix G.3 as an example). In this setting, the best accuracy is obtained with a smaller beta, while higher values result in less informative but more robust representations (Figure 3).

2) If the two views are complementary, containing mostly unrelated information, then there's probably little to be gained by treating them as separate views. In this scenario, we can just as well concatenate them into a single view and treat them accordingly, by, for example, applying the single-view data augmentation version of our method to capture known invariances or symmetries of the specific task.
We clarified this by adding a paragraph in our conclusive discussion.


Shared Question 2: Multi-View Information Bottleneck with more than 2 views
(Addresses Reviewer 2's 6th question and Reviewer 3's first question)

We believe our method can be extended to handle more than two views, but such an extension cannot be done trivially and so we have left it to future work.  Specifically, the mutual redundancy condition introduced in this work is not generally transitive (example in Appendix D). As a consequence, the sufficiency guarantee implied by Theorem B.2 does easily generalize for an arbitrary number of views, even when the mutual redundancy condition is respected by each pair of them.  Thus extending to more views requires considering more restrictive assumptions that take into account higher-order interactions between the different views and the label.

---

### Public Comment · ~Xuefeng_Du1 · 2020-02-27
**Questions with respect to the paired input data**

Hi, recently I read this paper and I found the idea really appealing but I have some concerns with respect to the "unsupervised setting" proposed in the paper. It seems like we have to input two images within the same class to the model (claimed in the beginning of section 3) but in that we are in the "unsupervised" setting, how can we obtain the paired labeled data without knowing the labels, especially when we are dealing with large-scale dataset.

Any suggestions will be appreciated.

Thanks,

---

> ### Author Response · Authors · 2020-03-02
> **Obtaining paired input data**
>
> Hi, thank you for expressing interest in our research and for the insightful question.
> The unsupervised multi-view setting approached in this paper refers to scenarios in which two redundant sources of information are accessible, while the target label is not.
>
> Examples of datasets that fit into this framework include pictures obtained with multi-lens cameras, audio signals from different microphones in the same room, consecutive frames in temporally consistent videos and multi-lingual corpora amongst many others, which have been described in previous multi-view learning literature [Xu 2013, Li 2018]
>
> Even when redundant sources of information are not available, it is possible to exploit known properties of the down-stream task and apply independent data augmentations to produce a multi-view dataset without having access to any label information (as visualized in Figure 1 and described in section 3.3 and 5.2).
>
> For this reason, we believe the unsupervised multi-view setting described in this work does not impose any fundamental restriction on the standard unsupervised settings, even if defining meaningful data augmentation strategies or identifying redundant sources of information can be challenging depending on the end-goal task and specific data-generating process.

---

### Decision · Program_Chairs · 2019-12-19

**Decision:**

Accept (Poster)

**Comment:**

This paper extends the information bottleneck method to the unsupervised representation learning under the multi-view assumption. The work couples the multi-view InfoMax principle with the information bottleneck principle to derive an objective which encourages the representations to contain only the information shared by both views and thus eliminate the effect of independent factors of variations. Recent advances in estimating lower-bounds on mutual information are applied to perform approximate optimisation in practice. The authors empirically validate the proposed approach in two standard multi-view settings.
Overall, the reviewers found the presentation clear, and the paper well written and well motivated. The issues raised by the reviewers were addressed in the rebuttal and we feel that the work is well suited for ICLR. We ask the authors to carefully integrate the detailed comments from the reviewers into the manuscript. Finally, the work should investigate and briefly establish a connection to [1].

[1] Wang et al. "Deep Multi-view Information Bottleneck". International Conference on Data Mining 2019 (https://epubs.siam.org/doi/pdf/10.1137/1.9781611975673.5)